# GFMate: Empowering Graph Foundation Models with Test-time Prompt Tuning

Yan Jiang [1]   Ruihong Qiu [1]   Zi Huang [1]

## Abstract

Graph prompt tuning has shown great potential in graph learning by introducing trainable prompts to enhance the model performance in conventional single-domain scenarios. Recent research has extended graph prompts to improve Graph Foundation Models (GFMs) by few-shot tuning auxiliary prompts. Despite their progress, most existing methods embed source-domain information into prompts, which serve either as input to GFMs or encoded during model pre-training. Such prompt entanglement with specific source domains and GFM pre-training strategy restricts their generalisability to other domains and different GFMs. Furthermore, existing GFM prompts merely rely on few-shot tuning for adaptation, neglecting the rich information in unlabelled target domain test data. Motivated by these insights, this paper aims to empower **GFM**s with pre-training-**a**gnostic **te**st-time graph prompt tuning, named **GFMate**. GFMate introduces centroid and layer prompts applied after pre-training on target domains, avoiding entanglement with specific source domains and model pre-training. In addition, a test-time complementary learning objective is devised to exploit both labelled and unlabelled target domain data for effective test-time prompt tuning. Extensive experiments on 12 benchmark datasets demonstrate the superior performance and efficiency of GFMate, achieving improvements of up to 30.63%. Code is available at https://github.com/YanJiangJerry/GFMate.

## 1. Introduction

Graph prompt tuning has demonstrated notable potential in single-domain graph supervised learning (Liu et al., 2023b; Fang et al., 2024; Zi et al., 2024; Sun et al., 2023; Chen

[1] School of Electrical Engineering and Computer Science, The University of Queensland, Brisbane, Queensland, Australia . Correspondence to: Yan Jiang <yan.jiang@uq.edu.au>.

*Proceedings of the $43^{rd}$ International Conference on Machine Learning*, Seoul, South Korea. PMLR 306, 2026. Copyright 2026 by the author(s).

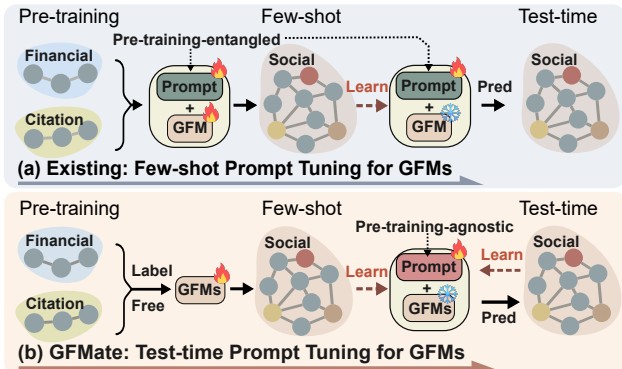

*Figure 1.* **Comparison between GFMate and existing GFM prompts.** (a) Existing prompts entangle with specific source domains and model pre-training strategies and adapt to target domains merely by few shots. (b) GFMate instead proposes pre-training-agnostic test-time prompts by learning from all target-domain data.

et al., 2025), where auxiliary prompts are supervised tuned to improve the performance of backbone models, such as GPPT (Sun et al., 2022) and ProNoG (Yu et al., 2024b). These prompts are typically designed as additive or multiplicative, learnable vectors on input features or encoded embeddings and are supervised trained within a single domain. Recent developments have further extended prompts to Graph Foundation Models (GFMs) (Zhao et al., 2024a), such as MDGPT (Yu et al., 2024c), SAMGPT (Yu et al., 2025) and MDGFM (Wang et al., 2025). Generally, prompts are jointly pre-trained with the GFM on source domains (e.g., financial network and citation network), and subsequently fine-tuned with few-shot samples on an unseen target domain (e.g., social network). This cross-domain scenario greatly enhances the transferability of graph learning models, which is the focus of this paper.

Unlike single-domain supervised graph prompting, applying prompts in cross-domain GFMs, where the target domain is unseen during pre-training, faces two primary challenges:

**(i) Existing GFM prompt designs are generally pre-training–entangled on a specific set of source domains and could face difficulty in generalising to unseen target domains**. For example, MDGFM (Wang et al., 2025) injects domain tokens into source domain graphs during pre-training, while SAMGPT (Yu et al., 2025) and MDGPT (Yu et al., 2024c) incorporate domain-related prompt vectors into the encoding process of GFM during pre-training. These GFM prompts are entangled with a specific set of

source-domain graphs, as illustrated in Figure 1 (a). However, domain-specific information learnt on the prompts cannot be straightforwardly transferred from the source domains to the target domains, as the target domain is unseen in GFM cross-domain scenarios, resulting in substantiate different graph structural and feature distribution (Zhao et al., 2024a; Jin et al., 2023). Additionally, these pre-training–entangled prompts cannot be easily generalised across different pre-trained models. For instance, the prompts in MDGFM (Wang et al., 2025) are jointly pre-trained with GFMs by a specifically designed graph contrastive learning conditioned on the source domain graph structure and cannot be applied to GFMs pre-trained with other strategies (e.g., link prediction).

Moreover, **(ii) current graph prompt tuning paradigm relies solely on few shots while neglecting abundant unlabelled samples despite their availability.** In the widely applied few-shot scenarios, GFM prompt vectors are optimised on a target domain graph where few-shot labelled nodes and unlabelled test nodes both exist. As illustrated in Figure 1 (a), the abundant unlabelled nodes from the target domain are available during prompt tuning but only being exploited passively as neighbouring contexts during message passing (Sun et al., 2022; Liu et al., 2023b; Fang et al., 2024; Chen et al., 2025; Yu et al., 2024b; Sun et al., 2023; 2025; Yu et al., 2024c; 2025). This leaves the distribution shift between the limited few-shot data and the abundant unlabelled test data unaddressed, preventing the GFM prompts from adequately capturing target-domain knowledge.

In light of the above observations, designing GFM prompts for cross-domain scenarios should satisfy two key properties: (i) the prompt should be **pre-training-agnostic**, placing emphasis on the GFM downstream stage without relying on prior assumptions about source domains or specific pre-training strategies; (ii) the **rich information contained in target-domain test data** should be effectively exploited to learn the prompt, thereby enabling improved GFM adaptation to the unseen target domains. To this end, a novel GFMate framework is proposed in this paper. GFMate introduces a centroid prompt and a layer prompt only after pre-training to achieve generalisability across domains and different GFMs. To further exploit the rich information in both few-shot labelled training nodes and unlabelled testing nodes in target domains for GFM prompt tuning, a test-time graph complementary learning objective is proposed. Extensive experiments on node and graph classifications across 12 datasets from diverse domains demonstrate the superior performance and efficiency of GFMate, achieving improvements of up to **30.63%**. Our contributions are:

- A novel **pre-training-agnostic prompt paradigm** is proposed to enhance GFMs' generalisability, allowing the prompts to be generalised across unseen target domains.

- A **test-time graph complementary learning** objective is proposed to exploit both labelled and unlabelled target domain data for effective test-time GFM prompt tuning.
- Extensive experiments across 12 benchmarks verify that GFMate **achieves state-of-the-art performance with significant efficiency and generalisability gains.**

## 2. Preliminary

A graph is denoted as $G = (\mathcal{V}, \mathcal{E})$, where $\mathcal{V}$ and $\mathcal{E}$ are sets of nodes and edges, which can also be represented as the feature $\boldsymbol{X} \in \mathbb{R}^{N \times d}$ and adjacency matrix $\boldsymbol{A} \in \mathbb{R}^{N \times N}$, where $N$ is the number of nodes and $d$ is the feature dimension. A graph foundation model is pre-trained on source domain graphs and adapted to target domains. Generally, target domains are unseen during pre-training.

**Few-shot Classification.** Most GFMs focus on few-shot node and graph classification (Zhao et al., 2024a; Yu et al., 2024c; 2025; Wang et al., 2025; Yuan et al., 2025). In few-shot node classification, each node $v \in \mathcal{V}$ is assigned a class $y \in \mathcal{Y}$, where $\mathcal{Y}$ is the set of node classes. For graph classification over a set of graphs $\mathcal{G}$, each graph $G \in \mathcal{G}$ receives a label $y \in \mathcal{Y}$, where $\mathcal{Y}$ denotes the graph-level classes. An $m$-shot task provides $m$ labelled data per class.

---

**Definition 1: Prompt Tuning for GFMs in Few-shot Classification. (See Figure 1 (a))**

Given a pre-trained $\text{GFM}_{\boldsymbol{\theta}^*}$ and associated prompts $\mathcal{B}^*_{\text{Pre}}$ optimised with a set of source training graphs $\mathcal{G}_{\text{Pre}}$ on task $\mathcal{L}_{\text{Pre}}$, current GFM prompt tuning aims to minimise the downstream task loss $\mathcal{L}_{\text{Fs}}$ by tuning the prompts $\mathcal{B}_{\text{Fs}}$ using only few shots $(\mathcal{V}_{\text{Fs}}, \mathcal{Y}_{\text{Fs}})$ from a target domain graph $\mathcal{G}_{\text{Tar}}$. The objective can be defined as:

$$\arg \min_{\mathcal{B}_{\text{Fs}}} \mathcal{L}_{\text{Fs}} \left( \text{GFM}_{\boldsymbol{\theta}^*}, \mathcal{B}_{\text{Fs}}, \mathcal{V}_{\text{Fs}}, \mathcal{Y}_{\text{Fs}} \right), \text{ s.t. } \mathcal{B}_{\text{Fs}} \leftarrow \mathcal{B}^*_{\text{Pre}}$$
(1)

---

**Remark 1: Pre-training-entangled GFM Prompts.** Generally, prompts are typically additive or multiplicative vectors applied to nodes or embeddings (Liu et al., 2023b; Chen et al., 2025; Yu et al., 2024b; Wang et al., 2025; Yu et al., 2024c; 2025), e.g., in GCOPE (Zhao et al., 2024a), $\boldsymbol{x}' = \boldsymbol{x} + \boldsymbol{b}$, while in SAMGPT (Yu et al., 2025), $\boldsymbol{h}' = \boldsymbol{h} \odot \boldsymbol{b}$, where $\odot$ denotes element-wise multiplication. Most existing methods learn prompts on source domains during pre-training and reuse or fine-tune them by the few-shot from target domain (Wang et al., 2025; Yu et al., 2024c; 2025), which corresponds to using $\mathcal{B}^*_{\text{Pre}}$ to initialise or replace $\mathcal{B}_{\text{Fs}}$ where $\mathcal{B}^*_{\text{Pre}}$ is optimised with the GFM by a pre-training loss $\mathcal{L}_{\text{Pre}}$. Such prompts are entangled with a specific set of source domains and pre-training strategy, limiting their generalisability to arbitrary target domains.

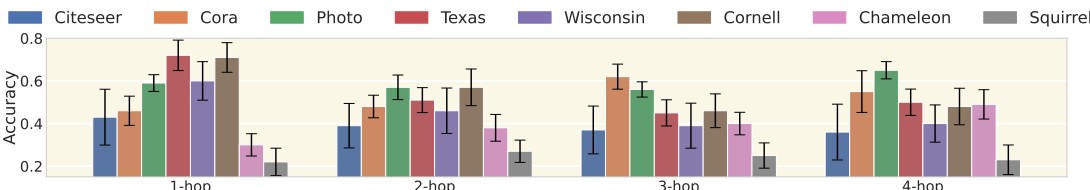

*Figure 2.* **Performance of the embeddings extracted from each layer of pre-trained GFMs varies greatly across different domains.**

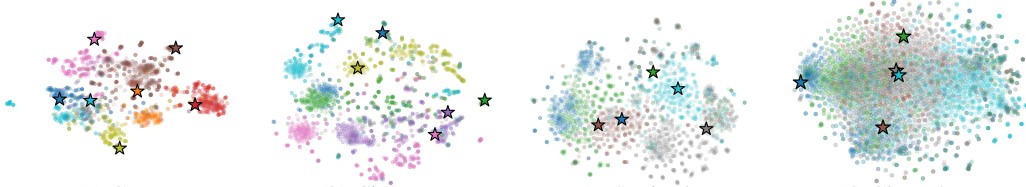

| (a) Cora. | (b) Citeseer. | (c) Squirrel. | (d) Chameleon. |

*Figure 3.* **Target domain node embedding visualisation by t-SNE.** Stars denote one-shot embeddings from GFM prompt method SAMGPT, which do not align with the test distribution. The GFM is pre-trained on all domains except the one for the target domain.

**Remark 2: Passive Utilisation of Unlabelled Nodes in GFM Prompts.** Although few-shot learning for graph prompt tuning typically and only utilises the supervision signals from the few labelled nodes, the unlabelled nodes in the target graph $\mathcal{G}_{\text{Tar}}$ are still required to be accessible to provide neighbourhood information for the labelled nodes (Wang et al., 2025; Yu et al., 2024c; 2025).

**Limitations of Pre-training-entangled Prompts**: Existing GFM prompts face several limitations for cross-domain:

**(i) Limited cross-domain generalisability.** Existing prompts generally assume the source domain shares a similar distribution with the target domain. This assumption may fail when the unseen target domain has a different distribution. For instance, MDGPT (Yu et al., 2024c) pre-trains source tokens based on this assumption, which may not generalise well to unseen target domains.

**(ii) Limited cross-model generalisability.** Existing prompts are often coupled with specific model architectures and pre-training strategies. For example, SAMGPT's (Yu et al., 2025) contrastive-based prompts are incompatible with models pre-trained by link prediction tasks.

**(iii) Sensitivity to source-domain imbalance.** Existing GFM prompts are pre-trained on multiple source domains. During pre-training, dominant source domains could suppress learning from others, leading to over-fitting and limiting generalisation to other domains. In contrast, a pre-training-agnostic prompt avoids such source-domain dominance and thus reduces this risk.

**(iv) Neglect of unlabelled target domain data.** Existing GFM prompts apply conventional few-shot fine-tuning (Liu et al., 2023b) but ignore abundant unlabelled data which are also available (Jin et al., 2023). This leaves the train-test distribution shift unresolved and hinders adaptation.

**Empirical Observations**: Two key issues arise during GFM downstream stage on target domains:

**(i) Hop-aggregation performance variation.** Target graphs exhibit distinct node neighbourhood patterns, causing performance variations. As in Figure 2, existing pre-training-entangled prompts fail to account for these data-level variations. Thus, pre-training-agnostic prompts that prioritise learning from the target domain are urgently needed.

**(ii) Train-test distribution shift.** GFM methods like SAMGPT (Yu et al., 2025) classify nodes via similarity with few-shot embeddings (Figure 3). However, pre-training-entangled prompts often yield unaligned embeddings between few-shot and test samples in unseen domains, causing classification errors. Furthermore, few-shot fine-tuning is sensitive to distribution shifts, as limited shots capture the distribution inadequately. Learning pre-training-agnostic prompts from both few-shot and test data can better capture the underlying distribution and improve performance.

More detailed discussions are provided in Appendix G and H. **These observations directly motivate our method design of pre-training-agnostic prompts on the target domain and the test-time learning objectives to capture the test distribution for effective prompt tuning.**

## 3. Proposed Method: GFMate

GFMate in Figure 4 focus on the following problems:

> **Definition 2: Test-time Prompt Tuning for GFMs in Few-shot Classification. (See Figure 1 (b))**
>
> Given a pre-trained $\text{GFM}_{\theta^*}$ optimised on source graphs $\mathcal{G}_{\text{Pre}}$ via task $\mathcal{L}_{\text{Pre}}$, test-time prompt tuning aims to minimise the downstream loss $\mathcal{L}_{\text{Te}}$ using a set of learnable prompts $\mathcal{B}_{\text{Te}}$. Unlike conventional methods, both few shots $(\mathcal{V}_{\text{Fs}}, \mathcal{y}_{\text{Fs}})$ and unlabelled samples $\mathcal{V}_{\text{Tar}} \setminus \mathcal{V}_{\text{Fs}}$ from the target domain graph $\mathcal{G}_{\text{Tar}}$ are exploited during test-time adaptation. The objective is defined as:
>
> $$\arg\min_{\mathcal{B}_{\text{Te}}} \mathcal{L}_{\text{Te}} \left( \text{GFM}_{\theta^*}, \mathcal{B}_{\text{Te}}, \mathcal{V}_{\text{Fs}}, \mathcal{y}_{\text{Fs}}, \mathcal{V}_{\text{Tar}} \setminus \mathcal{V}_{\text{Fs}} \right) \quad (2)$$

**Remark 3: Pre-training-agnostic GFM Prompts.** The prompt initialisation and training are unnecessary during pre-training, unlike existing approaches under Definition 1. The prompt relies on no prior assumption about the pre-trained model or strategies and is not constrained by the specific pre-training domain set, achieving better generalisability across domains and pre-trained GFMs.

**Remark 4: Active Exploitation of Unlabelled Nodes.** The unlabelled testing nodes are utilised to optimise the prompts with explicit learning strategies compared with existing work under Definition 1. This active exploitation of target domain testing data effectively mitigates the train-test distribution shift between the limited few-shot labelled nodes and the abundant testing nodes.

### 3.1. Label-free Cross-domain Pre-training

GFMate does not require a specifically customised GFM architecture or pre-training strategies. In this work, a general and straightforward GFM is employed, built on standard GNN backbones such as GCN (Kipf & Welling, 2017) and GAT (Veličković et al., 2018). During pre-training, following prior GFMs (Zhao et al., 2024a; Yu et al., 2025) under a consistent feature projection, singular value decomposition (SVD) is adopted to align node features into a latent space with unified dimensions. The model can then be pre-trained in a label-free manner by various self-supervised objectives, such as link prediction (Lu et al., 2021), graph contrastive learning (You et al., 2020) and deep graph infomax (Velickovic et al., 2019). Unless otherwise specified, the GFM is pre-trained by link prediction (Lu et al., 2021), eliminating the need for extensive prompt tuning during pre-training compared to existing GFM prompt methods (Yu et al., 2025; 2024c; Wang et al., 2025; Zhao et al., 2024a).

### 3.2. Centroids for Few-shot Classification

To perform downstream classification, centroids, which indicate the cluster centres of different classes of nodes, are used to compute similarities with testing samples and obtain classification results. Assume that the pre-trained GFM includes an $L$-layer GNN with optimal fixed parameters $\theta^*$ in hidden dimension $d$. At layer $l$, the target domain graph node embedding matrix is defined as $\boldsymbol{H}^{(l)} \in \mathbb{R}^{N \times d}$ where $\boldsymbol{H}^{(l)} = \text{GFM}_{\theta^*}^{(l)}(\boldsymbol{X}, \boldsymbol{A})$. The embedding of few-shot nodes is denoted by $\boldsymbol{h}_{\text{Fs}}^{(l)} \in \mathbb{R}^d$, and the embedding of testing nodes is denoted by $\boldsymbol{h}_{\text{Te}}^{(l)} \in \mathbb{R}^d$. The centroid $\boldsymbol{e}_c^{(l)} \in \mathbb{R}^d$ for class $c$ at layer $l$ can be initialised as the mean embedding of the few shots in class $c$, defined as:

$$\boldsymbol{e}_c^{(l)} = \frac{1}{|\mathcal{V}_{\text{Fs},c}|} \sum_{i \in \mathcal{V}_{\text{Fs},c}} \boldsymbol{h}_{\text{Fs},i}^{(l)}. \tag{3}$$

The centroid matrix $\boldsymbol{E}$ collects all centroids $\boldsymbol{e}_c^{(l)}$ across layers $l$ and classes $c$, with $\boldsymbol{E} \in \mathbb{R}^{L \times C \times d}$, and is used to compute embedding similarities for classification: $\hat{y}_i =$

$\arg\max_c \text{sim}(\boldsymbol{h}_i^{(l)}, \boldsymbol{E}^{(l)})$ where sim denotes cosine similarity function. To be noticed, it is slightly overloaded to represent similarity between a node embedding vector and each row vector from the centroid matrix.

### 3.3. Pre-training-agnostic Prompt Design

This section introduces GFMate prompt design. Existing methods construct prompts by adding to or multiplying input graph features or node embeddings within GFMs, which are tightly entangled with specific GFM pre-training loss and source domains. Motivated by empirical observation in Section 2, GFMate develops pre-training-agnostic prompts that can effectively enhance GFM downstream adaptation without prior assumptions about source domains or pre-training strategies. The key idea is to avoid interfering with node embeddings trained during pre-training and instead focus on test-time prediction on the target domain. Specifically, GFMate constructs two pre-training-agnostic prompts: the centroid prompt and the layer prompt $\mathcal{B}_{\text{Te}} = (\boldsymbol{\beta}, \boldsymbol{\eta})$.

**Centroid Prompts for Target-domain Centroid Movement.** The centroid prompts are designed to adapt the centroid representations for improved test-time performance on the target domain. The intuition is to adjust the centroids derived from few-shot nodes toward directions that better align with the classification task in the unseen target domain. Let $\boldsymbol{\beta}_c^{(l)} \in \mathbb{R}^d$ denote a $d$-dimension learnable prompt for the centroid of class $c$ at layer $l$, which is randomly initialised and added element-wise to the centroid $\boldsymbol{e}_c^{(l)}$, resulting in an refined centroid $\widetilde{\boldsymbol{e}}_c^{(l)}$, which can be defined as:

$$\widetilde{\boldsymbol{e}}_c^{(l)} = \boldsymbol{e}_c^{(l)} + \boldsymbol{\beta}_c^{(l)}. \tag{4}$$

With the centroids initialised by few-shot samples, the additive centroid prompt $\boldsymbol{\beta}$ provides possibility for centroids to move towards the real centre of a cluster in target domains.

**Layer Prompts for Multi-layer Ensemble Prediction.** During test-time, the performance of neighbourhood aggregation across different hops in a pre-trained GFM varies substantially between target domains as observed in Section 2, leading to inconsistent layer-wise prediction accuracy. To adapt a pre-trained GFM to an arbitrary target domain, the predictions from all layers are ensembled by a layer prompt $\boldsymbol{\eta} \in \mathbb{R}^L$, which dynamically adjusts the contribution of each layer to exploit hop-aggregation patterns across different target domains. Specifically, given the refined centroid matrix by the centroid prompts for all classes at layer $l$, $\widetilde{\boldsymbol{E}}^{(l)}$, the final multi-layer ensemble prediction $\hat{y}$ for node $i$ in an unseen target domain is defined by the class with the maximum ensemble probability:

$$\hat{y}_i = \arg\max_c \left[ \text{Softmax}\left( \sum_{l=0}^{L} \eta^{(l)} \cdot \text{sim}(\boldsymbol{h}_i^{(l)}, \widetilde{\boldsymbol{E}}^{(l)}) \right) \right]. \tag{5}$$

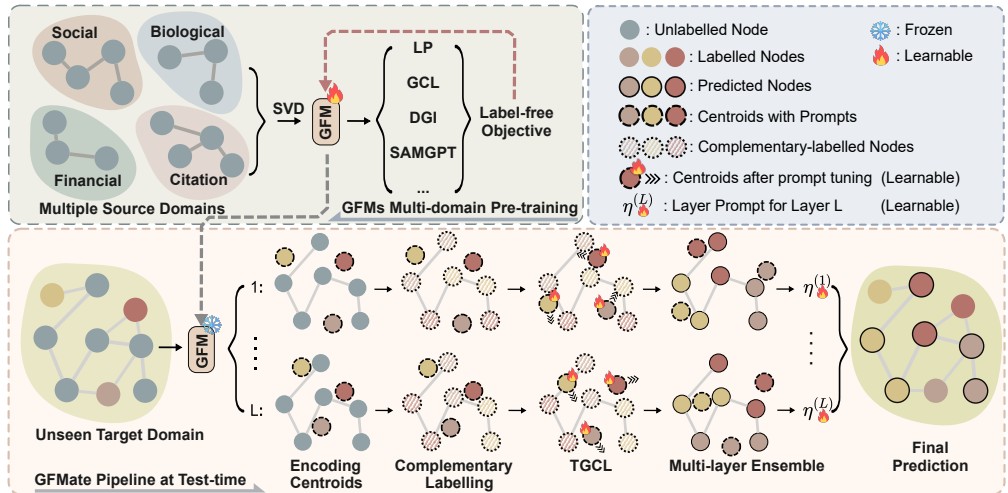

*Figure 4.* **The overall framework for GFMate.** The fixed GFMs can be pre-trained with self-supervised objectives in a label-free manner on source domains. GFMate is then applied at test-time to tune the pre-training-agnostic prompts, specifically the centroid prompts and the layer prompts, enabling the GFM to achieve better prediction results on unseen target domain graphs.

The layer prompt $\boldsymbol{\eta}$ allows our method to **adaptively learn from the hop-aggregation patterns of the target domain**, effectively ensembling the layer-wise predictions corresponding to different hop-aggregated representations to improve the GFM adaptation on the unseen target domain.

### 3.4. Test-time Graph Complementary Learning

During test-time, directly tuning prompts with pseudo labels of the test data often leads to severe degradation due to inaccuracy (Appendix E.7). To address this, GFMate introduces test-time graph complementary learning (TGCL) to jointly trains on few shots and complementary-labelled test nodes. The intuition is to learn from the predicted least similar class $\overline{y}$. Since prediction performance varies across layers for target domains, complementary labels are derived by a layer-wise entropy-based strategy. Specifically, the layer-wise entropy score $H_i^{(l)}$ for test node $i$ is:

$$
\begin{aligned}
H_i^{(l)} &= -\sum_{c=1}^{C} p_{i,c}^{(l)} \log p_{i,c}^{(l)}, \\
p_i^{(l)} &= \mathrm{Softmax}(\mathrm{sim}(\boldsymbol{h}_i^{(l)}, \boldsymbol{E}^{(l)})).
\end{aligned}
\tag{6}
$$

The layer with the lowest average entropy score on the target domain graph, indicating the highest prediction confidence, is selected as the pivot layer $\hat{l}$. Then, the complementary label $\overline{y}$ is defined as the class with the lowest similarity at the pivot layer: $\overline{y}_i = \arg\min_c \mathrm{sim}(\boldsymbol{h}_i^{(\hat{l})}, \boldsymbol{E}^{(\hat{l})})$, which is evaluated in the Appendix E.8 in comparison to the pseudo-labels. Once all testing nodes are complementary-labelled, the test-time learning loss $\mathcal{L}_{\mathrm{Te}}$ is defined as:

$$
\begin{aligned}
\mathcal{L}_{\mathrm{Te}} &= -\sum_{l=0}^{L} \frac{1}{|\mathcal{V}_{\mathrm{Te}}|} \sum_{v_i \in \mathcal{V}_{\mathrm{Te}}} \log\left(1 - p_{\overline{y}_i}^{(l)}\right), \\
\text{where } p_{\overline{y}_i}^{(l)} &= \frac{\exp\left(\eta^{(l)} \cdot \mathrm{sim}(\boldsymbol{h}_i^{(l)}, \widetilde{\boldsymbol{e}}_{\overline{y}_i}^{(l)})/\tau\right)}{\sum_{c=1}^{C} \exp\left(\eta^{(l)} \cdot \mathrm{sim}(\boldsymbol{h}_i^{(l)}, \widetilde{\boldsymbol{e}}_c^{(l)})/\tau\right)}
\end{aligned}
\tag{7}
$$

where $\boldsymbol{\eta}$ is the learnable layer prompt and $\tau$ is a temperature that controls the smoothness of the probability distribution. $\mathcal{V}_{\mathrm{Te}}$ refers to the testing nodes with complementary labels.

Intuitively, the test-time learning loss **encourages centroids to be distant from testing samples being predicted to the most dissimilar class** (the complementary class $\overline{y}$), which in turn promotes increased similarity between centroids and testing samples from the other more similar classes. To evaluate the generalisation capability of the proposed test-time learning loss on complementary-labelled testing nodes, an excess risk bound is established:

---

**Proposition 1 (Excess Risk Bound for Test-time Learning in GFMate.)**

Let $\mathcal{F}$ be the hypothesis space, where each predictor model $f \in \mathcal{F}$ refers to a GFM with prompts, defined as $f = (\mathrm{GFM}_{\theta^*}, \mathcal{B})$ with pre-trained parameters $\theta^*$ and learnable prompts $\mathcal{B}$. Let $\overline{\mathcal{L}}$ be the test-time learning loss over test data $x$ and complementary labels $\overline{y}$. The population risk is defined as:

$$
R_{\overline{\mathcal{L}}}(f) = \mathbb{E}_{(x,\overline{y})}\left[\overline{\mathcal{L}}(f(x), \overline{y})\right].
$$

Define the empirical risk minimiser as $\hat{f} = \arg\min_{f \in \mathcal{F}} \hat{R}_{\overline{\mathcal{L}}}(f)$. With probability at least $1 - \delta$, the generalisation bound on the excess risk holds:

$$
R_{\overline{\mathcal{L}}}(\hat{f}) - \min_{f \in \mathcal{F}} R_{\overline{\mathcal{L}}}(f) \leq 4C\ell_\rho \mathfrak{R}(\mathcal{F}) + 2\sqrt{\frac{\log(1/\delta)}{2N}}.
$$

---

Here, $C$ denotes the number of classes, $\ell_\rho$ is the Lipschitz constant of the complementary loss function $\overline{\mathcal{L}}$, and $\mathfrak{R}(\mathcal{F})$ is the Rademacher complexity (Shalev-Shwartz & Ben-David, 2014) of the hypothesis class $\mathcal{F}$, which consists of all predictor models with the same GFM parameter

$\theta^*$ but varying prompts $\mathcal{B}$. $N = |\mathcal{V}_{\text{Te}}|$ is the number of complementary-labelled test samples, and $\delta$ is the confidence level. The proof and more detailed definition are deferred to Appendix D.

Intuitively, Proposition 1 suggests that a smaller number of classes or a greater number of complementary-labelled samples leads to a tighter risk upper bound of our GFM prompts. **In light of this insight, GFMate exploits all testing samples in the test-time complementary learning process. These theoretical insights are further validated by empirical experiments in Section 4.7 and Appendix E.2.**

Meanwhile, by complementing the test-time learning objective on the testing samples, a few-shot learning loss $\mathcal{L}_{\text{Fs}}$ is introduced to learn from the few-shot labelled samples $\mathcal{V}_{\text{Fs}}$:

$$
\mathcal{L}_{\text{Fs}} = -\sum_{l=0}^{L} \frac{1}{|\mathcal{V}_{\text{Fs}}|} \sum_{v_i \in \mathcal{V}_{\text{Fs}}} \log p_{y_i}^{(l)},
$$

$$
\text{where } p_{y_i}^{(l)} = \frac{\exp\left(\eta^{(l)} \cdot \text{sim}(\boldsymbol{h}_i^{(l)}, \widetilde{\boldsymbol{e}}_{y_i}^{(l)})/\tau\right)}{\sum_{c=1}^{C} \exp\left(\eta^{(l)} \cdot \text{sim}(\boldsymbol{h}_i^{(l)}, \widetilde{\boldsymbol{e}}_c^{(l)})/\tau\right)}. \tag{8}
$$

Intuitively, the few-shot learning loss leverages the few-shot supervision signals to guide prompt optimisation by **maximising the similarity between centroids and labelled data with the same class while pushing the centroids away from different class samples.** Together, the final TGCL objective is defined as a convex combination of the few-shot and test-time learning losses to exploit all data from the target domain, which is optimised as:

$$
\mathcal{L}_{\text{TGCL}} = \gamma \mathcal{L}_{\text{Te}} + (1-\gamma)\mathcal{L}_{\text{Fs}}, \tag{9}
$$

where $\gamma$ is a hyperparameter in range $(0,1)$ that regulates the relative contribution of each loss function. The centroid prompts and the layer prompts for layer $l$ can be then optimised by:

$$
\begin{aligned}
\boldsymbol{\beta}_c^{(l)} &= \boldsymbol{\beta}_c^{(l)} - \alpha\nabla_{\boldsymbol{\beta}_c^{(l)}}\mathcal{L}_{\text{TGCL}}(\boldsymbol{\beta}_c^{(l)}), \\
\eta^{(l)} &= \eta^{(l)} - \alpha\nabla_{\eta^{(l)}}\mathcal{L}_{\text{TGCL}}(\eta^{(l)}),
\end{aligned} \tag{10}
$$

where $\alpha$ is the learning rate. In this way, the centroid and layer prompts are jointly learned on both the few shots and the testing samples from the target domain, adequately capturing the target domain distribution, thereby facilitating effective GFM downstream adaptation.

### 3.5. Extend GFMate to Graph Classification

GFMate can be easily extended from node-level to graph-level classification by designing a subgraph classification task following (Zhao et al., 2024a). The subgraph embedding can be computed by averaging the node embeddings, while the subgraph label is determined by the central node. This process enables seamless integration of GFMate into both node- and graph-level classification tasks.

### 3.6. Complexity Analysis

The time complexity of GFMate consists of two components: (i) Test-time Graph Complementary Learning and (ii) Multi-layer Ensembling. Both require $\mathcal{O}(dLNC)$, where $N$ is the number of nodes, $E$ the number of edges, $L$ the number of model layers, $C$ the number of classes, and $d$ the hidden dimension. The overall complexity is $\mathcal{O}(dL(N+E+NC))$. In practice, because $C \ll d$, this can be approximated as linear in the target domain graph size $\mathcal{O}(dL(N+E))$.

## 4. Experiments

**Datasets.** Twelve graph benchmark datasets across diverse domains are adopted, varying in size up to one hundred thousand nodes and millions of edges. For **node classification**: (i) Social Network: Cornell, Texas, Wisconsin, Chameleon, and Squirrel, where nodes represent social entities; (ii) Citation Network: Cora, Citeseer, and Arxiv-year, where nodes denote academic papers and edges indicate citation relationships; (iii) Commercial System: Amazon-photo, where nodes represent products and edges denote co-purchase relationships. For **graph classification**: (i) Biological Network: BZR, COX2, and PROTEINS, where nodes are biological entities and edges represent interactions. Detailed dataset statistics are provided in the Appendix B. The cross-domain GFM node classification setting follows a one-versus-all setting in GCOPE (Zhao et al., 2024a), where one dataset serves as the unseen target domain and all remaining datasets serve as the source domains. Graph classification maintains fairness in comparisons.

**Baselines.** Twenty-one baselines are compared with GFMate. These include: (i) **Single domain supervised learning (SL)** such as GCN (Kipf & Welling, 2017), GAT (Veličković et al., 2018), SAGE (Hamilton et al., 2017), H2GCN (Zhu et al., 2020a), and GPR (Chien et al., 2021); (ii) **Single domain self-supervised pre-training and fine-tuning (SSL + FT)**, including Link Prediction (LP) (Lu et al., 2021), DGI (Velickovic et al., 2019), and GCL (You et al., 2020). Fine-tuning (FT) is used to train the predictor; (iii) **Single domain pre-training and prompt tuning (SSL + Prompt)**, such as GPPT (Sun et al., 2022), All-In-One (Sun et al., 2023), ProNoG (Yu et al., 2024b), DAG-Prompt (Chen et al., 2025), GraphPrompt (Liu et al., 2023b), and GPF (Fang et al., 2024). (iv) **Cross domain GFM pre-training and prompt tuning (GFMs)**, including prompt-based GCOPE (Zhao et al., 2024a), MDGFM (Wang et al., 2025), MDGPT (Yu et al., 2024c), SAMGPT (Yu et al., 2025), BRIDGE (Yuan et al., 2025) and manifold-based RiemannGFM (Sun et al., 2025). GraphAny (Zhao et al., 2025) and GTrans (Jin et al., 2023) focus on full-shot settings in Appendix F. **LLM-based GFMs are not applicable since our datasets cover various non-text situations.** All baselines are categorised in Table 7 in the Appendix B.

*Table 1.* **Cross-domain transfer learning performance** of one-shot node classification. Results with different shots are in Appendix F. Average accuracy (%) over five runs is reported. The upper half shows single-domain training and testing setting, while the bottom half is cross-domain setting. ProNoG* refers to ProNoG that is implemented in a multi-domain pre-training and cross-domain testing setup (using SVD to unify dimensions) for a fair comparison with GFM methods. Results are marked as **best** and runner-up. OOM denotes out-of-memory on a 48 GB A6000 GPU, and "–" denotes that the official code has not been released for implementation on these datasets.

| | Methods | Texas | Cornell | Wisconsin | Chameleon | Squirrel | Arxiv-year | Cora | Citeseer | Photo |
|---|---|---|---|---|---|---|---|---|---|---|
| | | **Single-domain Training and Testing** | | | | | | | | |
| SL | GCN | $38.82_{\pm9.79}$ | $24.58_{\pm10.09}$ | $43.36_{\pm13.17}$ | $24.30_{\pm6.59}$ | $19.96_{\pm5.89}$ | $18.64_{\pm6.91}$ | $29.85_{\pm8.98}$ | $33.39_{\pm11.86}$ | $47.09_{\pm5.81}$ |
| | GAT | $39.96_{\pm9.62}$ | $25.84_{\pm12.26}$ | $45.99_{\pm10.86}$ | $22.81_{\pm7.38}$ | $20.77_{\pm4.48}$ | $19.93_{\pm5.40}$ | $33.25_{\pm9.72}$ | $35.51_{\pm9.70}$ | $47.33_{\pm4.97}$ |
| | SAGE | $40.38_{\pm8.21}$ | $32.55_{\pm13.36}$ | $47.41_{\pm11.36}$ | $31.29_{\pm8.87}$ | $22.34_{\pm4.32}$ | $20.75_{\pm4.99}$ | $35.76_{\pm8.89}$ | $38.80_{\pm9.73}$ | $49.72_{\pm3.81}$ |
| | GPR | $37.60_{\pm9.54}$ | $30.77_{\pm13.42}$ | $49.56_{\pm15.58}$ | $28.44_{\pm6.65}$ | $21.06_{\pm6.14}$ | $10.81_{\pm9.94}$ | $38.99_{\pm15.77}$ | $29.77_{\pm13.22}$ | $43.39_{\pm4.66}$ |
| | H2GCN | $47.75_{\pm12.89}$ | $35.58_{\pm15.02}$ | $45.55_{\pm10.61}$ | $28.01_{\pm9.50}$ | $21.10_{\pm3.06}$ | $15.54_{\pm7.33}$ | $30.90_{\pm9.98}$ | $30.91_{\pm12.79}$ | $45.81_{\pm6.09}$ |
| SSL+FT | LP+FT | $36.93_{\pm11.90}$ | $23.88_{\pm11.43}$ | $41.37_{\pm14.22}$ | $25.34_{\pm7.19}$ | $20.04_{\pm5.61}$ | $17.94_{\pm6.06}$ | $35.59_{\pm9.74}$ | $34.92_{\pm12.08}$ | $48.82_{\pm6.55}$ |
| | DGI+FT | $34.46_{\pm12.92}$ | $22.89_{\pm11.32}$ | $38.89_{\pm15.77}$ | $25.78_{\pm7.34}$ | $20.85_{\pm6.57}$ | $18.06_{\pm6.22}$ | $32.38_{\pm8.86}$ | $33.96_{\pm11.57}$ | $45.17_{\pm7.34}$ |
| | GCL+FT | $28.81_{\pm15.32}$ | $20.56_{\pm13.36}$ | $35.70_{\pm17.73}$ | $24.69_{\pm8.82}$ | $19.97_{\pm5.62}$ | $15.53_{\pm8.89}$ | $33.27_{\pm9.50}$ | $36.05_{\pm13.44}$ | $47.33_{\pm5.89}$ |
| SSL+Prompt | GPPT | $44.47_{\pm10.88}$ | $29.74_{\pm8.32}$ | $35.16_{\pm9.07}$ | $29.91_{\pm6.48}$ | $21.16_{\pm5.95}$ | $19.96_{\pm7.63}$ | $40.62_{\pm8.69}$ | $39.79_{\pm10.67}$ | $50.19_{\pm7.74}$ |
| | GPF | $47.34_{\pm12.72}$ | $52.15_{\pm14.53}$ | $40.19_{\pm14.49}$ | $30.95_{\pm9.18}$ | $22.71_{\pm4.87}$ | $20.89_{\pm8.20}$ | $45.75_{\pm9.61}$ | $40.51_{\pm12.79}$ | $49.38_{\pm6.56}$ |
| | ProNoG | $55.31_{\pm12.92}$ | $48.49_{\pm11.54}$ | $46.29_{\pm17.74}$ | $31.19_{\pm8.09}$ | $24.25_{\pm4.79}$ | OOM | $56.54_{\pm12.33}$ | $37.79_{\pm13.35}$ | $47.72_{\pm6.60}$ |
| | GraphPrompt | $53.27_{\pm9.95}$ | $55.13_{\pm13.39}$ | $45.03_{\pm11.83}$ | $33.29_{\pm9.19}$ | $23.02_{\pm4.89}$ | $22.61_{\pm4.66}$ | $49.77_{\pm8.82}$ | $38.69_{\pm13.98}$ | $46.65_{\pm6.53}$ |
| | DAGPrompt | $68.27_{\pm10.02}$ | $59.11_{\pm10.04}$ | $50.49_{\pm11.59}$ | $37.79_{\pm6.62}$ | $25.67_{\pm6.34}$ | $23.08_{\pm8.14}$ | $54.88_{\pm9.24}$ | $47.24_{\pm9.59}$ | $52.96_{\pm6.07}$ |
| | All-In-One | $63.79_{\pm15.91}$ | $57.24_{\pm12.70}$ | $55.35_{\pm18.36}$ | $27.94_{\pm6.31}$ | $21.18_{\pm7.06}$ | $15.29_{\pm7.55}$ | $49.92_{\pm11.75}$ | $40.69_{\pm15.88}$ | $52.25_{\pm7.33}$ |
| | | **Multi-domain Pre-training and Cross-domain Fine-tuning then Testing** | | | | | | | | |
| GFM Methods | ProNoG* | $48.25_{\pm9.60}$ | $37.18_{\pm10.04}$ | $39.70_{\pm12.32}$ | $26.48_{\pm8.69}$ | $22.10_{\pm5.52}$ | OOM | $40.07_{\pm13.35}$ | $35.56_{\pm10.63}$ | $45.28_{\pm7.02}$ |
| | GCOPE | $64.76_{\pm14.84}$ | $60.98_{\pm14.66}$ | $54.66_{\pm12.86}$ | $30.58_{\pm7.44}$ | $22.16_{\pm5.77}$ | $17.98_{\pm5.51}$ | $39.06_{\pm12.52}$ | $42.26_{\pm14.19}$ | $55.69_{\pm4.68}$ |
| | MDGPT | $59.76_{\pm12.44}$ | $54.19_{\pm13.08}$ | $50.40_{\pm15.07}$ | $28.04_{\pm4.28}$ | $24.41_{\pm7.01}$ | OOM | $44.52_{\pm11.39}$ | $41.98_{\pm12.24}$ | $54.96_{\pm10.25}$ |
| | SAMGPT | $66.79_{\pm10.77}$ | $59.34_{\pm9.82}$ | $52.29_{\pm14.40}$ | $38.12_{\pm8.90}$ | $25.75_{\pm6.29}$ | OOM | $52.83_{\pm12.04}$ | $47.76_{\pm10.55}$ | $56.33_{\pm9.04}$ |
| | MDGFM | – | $40.77_{\pm5.96}$ | – | $28.36_{\pm3.65}$ | $24.30_{\pm3.26}$ | – | $44.83_{\pm7.41}$ | $42.18_{\pm6.41}$ | – |
| | BRIDGE | $53.35_{\pm11.90}$ | $49.28_{\pm12.56}$ | $48.85_{\pm13.35}$ | $32.75_{\pm6.62}$ | $19.89_{\pm9.02}$ | $24.47_{\pm6.68}$ | $44.08_{\pm9.54}$ | $38.89_{\pm8.72}$ | $58.79_{\pm11.58}$ |
| | RiemannGFM | $58.60_{\pm15.27}$ | $46.35_{\pm11.92}$ | $47.32_{\pm16.58}$ | $29.68_{\pm9.95}$ | $20.13_{\pm8.58}$ | OOM | $37.91_{\pm16.13}$ | $38.02_{\pm9.58}$ | $49.69_{\pm13.32}$ |
| | **GFMate** | **$76.63_{\pm7.81}$** | **$79.67_{\pm8.47}$** | **$63.01_{\pm10.78}$** | **$47.25_{\pm6.11}$** | **$27.02_{\pm6.22}$** | **$30.19_{\pm3.65}$** | **$59.68_{\pm5.37}$** | **$56.25_{\pm13.33}$** | **$58.85_{\pm2.17}$** |

**Implementation.** For all methods, GCN is the default backbone model. Analyses of GFMate on GFMs by different backbones are presented in Section 4.6. For each target dataset, after obtaining the few-shot training samples, the remaining data are randomly split in a 1:9 ratio for validation and testing, following prior GFMs (Zhao et al., 2024a). This ensures a fair and consistent setting for all methods. More details, including hyperparameters and pseudo-codes, are provided in the reproducibility statement in Appendix B.

### 4.1. Overall Performance

The overall performance of GFMate is first evaluated on one-shot node classification tasks against all baselines. GFMate is integrated into GFMs with a default GCN backbone with LP pre-training. **Overall, GFMate outperforms all baselines across all domains.** (i) For multi-domain pre-training and cross-domain adaptation settings, in comparison with existing GFM-based methods, GFMate achieves the highest performance without requiring a delicate design of the underlying GFMs. Furthermore, GFMate prompts are pre-training-agnostic and do not require prior assumptions on source domains and GFM pre-training strategies. **To be noticed, directly extending the traditional single-domain graph prompts to the GFM cross-domain setting results** in suboptimal performance, as shown by ProNoG*. (ii) Compared with traditional single-domain methods where the source and target domains are the same, applying prompt tuning on top of supervised training can enhance performance, as observed in DAGPrompt and All-In-One.

More in-depth analyses are provided, structured as: **efficiency in Section 4.2 and Appendix E.1; GFMate with different backbones and pre-training strategies in Section 4.6 and Section 4.5; robustness under pre-training domain shifts, test-time distribution shifts and noisy cases in Appendix E.3 and E.4; hyperparameter sensitivity in Appendix E.6; different few-shot settings, additional baselines, and graph classification in Section 4.4 and Appendix F.1 and F.2, respectively.**

### 4.2. Efficiency Analysis

**GFMate is highly efficient in both the convergence time and the GPU memory usage.** To ensure a fair comparison, convergence time is measured only for downstream adaptation for all methods, after GFM pre-training has been completed and the target data are available. The prompts in GFMate are only integrated with centroids and the multi-layer predictions, offering a less complicated and lightweight design compared to existing methods that require learnable

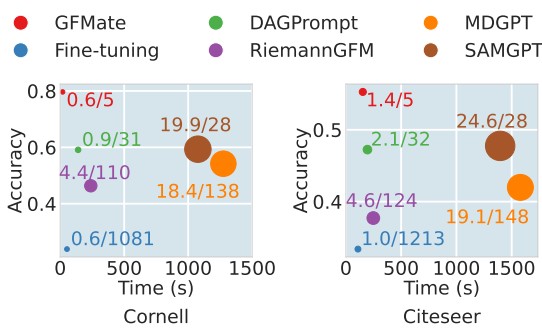

Figure 5. **Efficiency analysis.** The numbers in the figure indicate "GPU peak memory (k MB)/number of tunable parameters (k)". GFMate (red dot) has superior efficiency in time and memory while achieving higher performance.

prompts for each source domain and target domain data (Yu et al., 2024c; 2025). Furthermore, the substantially reduced number of tunable parameters leads to lower memory utilisation. The maximum improvements in time and memory efficiency reach **98.24%** and **97.18%**, respectively, over state-of-the-art methods SAMGPT. With such significant efficiency gains, GFMate emerges as the most effective approach among existing GFM-based methods, highlighting the superiority of its unique test-time learning design. More detailed statistics are provided in the Appendix E.1.

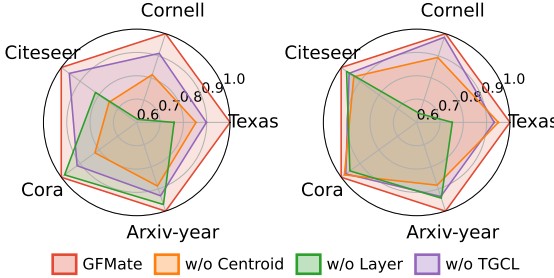

Figure 6. **Ablation studies.** One-shot (left) and three-shot (right) with cross-domain node classification results.

### 4.3. Ablation Study

**The ablation study verifies the effectiveness of each key component in GFMate.** In Figure 6, the three key modules, centroid prompt tuning (Centroid), layer prompt tuning (Layer), and test-time graph complementary learning (TGCL) are evaluated. "w/o Layer" denotes mean layer performance without the proposed multi-layer ensemble, and "w/o TGCL" denotes few-shot prompt tuning without the proposed test-time learning. The results verify that all three components contribute to GFMate's overall performance.

### 4.4. Effectiveness upon different Few-shot Scenarios

**With different numbers of labelled data (shots) in few-shot tuning, GFMate consistently outperforms other**

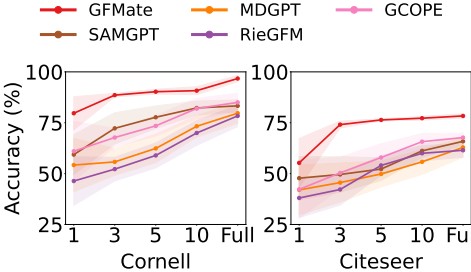

Figure 7. **GFMate in different few-shot settings by one-shot node classification.** GFMate consistently outperforms existing GFMs. RieGFM is RiemannGFM. The shaded regions indicate standard deviation. More results on more shots and other datasets are deferred to Appendix F due to page limit.

Table 2. **Plug-in GFMate on different pre-training methods under 1-shot setting.** Results under other shot settings are reported in Appendix Table 10.

|  | Texas | Cornell | Citeseer | Cora |
|---|---|---|---|---|
| **LP+FT** | $36.93_{\pm 11.90}$ | $23.88_{\pm 11.43}$ | $34.92_{\pm 12.08}$ | $35.59_{\pm 9.74}$ |
| **+GFMate** | $76.63_{\pm 7.81}$ | $79.67_{\pm 8.47}$ | $56.25_{\pm 13.33}$ | $59.68_{\pm 5.37}$ |
| **DGI+FT** | $34.46_{\pm 12.92}$ | $22.89_{\pm 11.32}$ | $33.96_{\pm 11.57}$ | $32.38_{\pm 8.86}$ |
| **+GFMate** | $74.80_{\pm 9.97}$ | $75.06_{\pm 10.37}$ | $52.31_{\pm 12.96}$ | $54.08_{\pm 9.04}$ |
| **GCL+FT** | $28.81_{\pm 15.32}$ | $20.56_{\pm 13.36}$ | $36.05_{\pm 13.44}$ | $33.27_{\pm 9.50}$ |
| **+GFMate** | $69.82_{\pm 11.36}$ | $73.29_{\pm 10.20}$ | $53.30_{\pm 13.19}$ | $58.76_{\pm 8.89}$ |
| **MDGPT** | $59.76_{\pm 12.44}$ | $54.19_{\pm 13.08}$ | $41.98_{\pm 12.24}$ | $44.52_{\pm 11.39}$ |
| **+GFMate** | $75.89_{\pm 9.60}$ | $75.47_{\pm 10.06}$ | $54.70_{\pm 10.62}$ | $57.20_{\pm 7.95}$ |
| **SAMGPT** | $66.79_{\pm 10.77}$ | $59.34_{\pm 9.82}$ | $47.76_{\pm 13.06}$ | $52.83_{\pm 12.04}$ |
| **+GFMate** | $72.54_{\pm 8.89}$ | $76.34_{\pm 9.17}$ | $55.73_{\pm 12.25}$ | $58.49_{\pm 7.93}$ |

**GFM methods.** As in Figure 7, GFMate is evaluated using 1, 3, 5, 10 and all labelled data for prompt tuning. GFMate, represented by the red line, consistently outperforms existing GFM methods with relatively small standard deviations, particularly on graph classification datasets. Even in the full-shot setting, where all labelled data are available for model tuning, GFMate remains consistently more effective than other GFM baselines.

### 4.5. On Generalisability of GFMate

**Applying GFMate on GFMs pre-trained with various methods can significantly enhance their performance.** As a pre-training-agnostic prompt tuning method, GFMate can be seamlessly integrated with GFMs pre-trained by different objectives. In Table 2, with SSL-based pre-trained models like LP, DGI and GCL, GFMate can increase the performance over the common fine-tuning (FT) under the 1-shot setting. The GFMs from MDGPT and SAMGPT are pre-trained using specifically designed link prediction and GCL objectives, with prompts injected into the pre-training process. It is clear that applying GFMate to these various pre-trained GFMs leads to significant performance improvements, regardless of their pre-training strategies and backbone models. Results under other shot settings are re-

ported in Appendix Table 10. These results demonstrate the strong plug-in ability and generalisability of the proposed pre-training-agnostic prompts.

*Table 3.* **Effectiveness on GNN-based GFMs with different backbones under 1-shot setting.** The default backbone is GCN, following prior work (Yu et al., 2025; 2024c; Wang et al., 2025). Results under other shot settings are reported in Appendix Table 9.

| | Texas | Cornell | Citeseer | Cora |
|---|---|---|---|---|
| **GFMate** | $76.63_{\pm 7.81}$ | $79.67_{\pm 8.47}$ | $56.25_{\pm 13.33}$ | $59.68_{\pm 5.37}$ |
| **w/ SAGE** | $78.19_{\pm 8.82}$ | $80.05_{\pm 8.24}$ | $57.39_{\pm 11.40}$ | $60.08_{\pm 5.37}$ |
| **w/ GAT** | $77.14_{\pm 7.42}$ | $79.58_{\pm 8.41}$ | $54.19_{\pm 13.50}$ | $59.30_{\pm 6.11}$ |
| **w/ H2GCN** | $82.31_{\pm 9.92}$ | $81.89_{\pm 9.43}$ | $52.38_{\pm 15.52}$ | $58.75_{\pm 5.64}$ |

### 4.6. GFMate with Different Backbone Models

In this section, GFMate is plugged into GFMs implemented with different GNN backbone models, including SAGE, GAT, and H2GCN. The default pre-training strategy is link prediction. As shown in Table 3, GFMate effectively improves GFMs across different backbones under the 1-shot setting, with GFMate applied to the GCN backbone consistently achieving strong performance across all datasets. Results under other shot settings are reported in Appendix Table 9. These results demonstrate the effectiveness of GFMate on diverse GFM architectures.

### 4.7. GFMate in Binary Classification Cases

The effectiveness of GFMate is further evaluated on binary classification cases. To establish a balanced classification setting, the datasets were modified as follows:

- For the **Cora** and **Amazon-Photo** datasets, four randomly selected classes were merged into a single group, with the remaining three classes treated as the second group.

- For the **Citeseer** and **Chameleon** datasets, three randomly selected classes were merged into one group, and the remaining classes constituted the second group.

This process successfully established a two-class, or binary, setting for subsequent evaluation. The performance comparison against the current state-of-the-art baseline, SAMGPT, is presented in Table 4. The results clearly demonstrate that GFMate achieves a significant performance advantage over the SOTA baseline. Crucially, the performance improvement achieved by GFMate over the baseline in the binary classification setting is consistently larger than the improvement observed in the multi-class classification setting. **This enhanced result in the binary case aligns with the theoretical underpinnings of Proposition 1, suggesting that fewer classes lead to improved generalisation.**

*Table 4.* **Comparison of performance gain of GFMate with SAMGPT on binary classification.**

| Methods | Chameleon (5) | Chameleon (2) | Cora (7) | Cora (2) |
|---|---|---|---|---|
| SAMGPT | 38.12±8.90 | 60.61±10.36 | 52.83±12.04 | 71.19±7.80 |
| GFMate | **47.25±6.11** | **78.88±7.56** | **59.68±5.37** | **81.77±3.51** |
| △ | 9.13 | 18.27 | 6.85 | 10.58 |

| Methods | Citeseer (6) | Citeseer (2) | Photo (8) | Photo (2) |
|---|---|---|---|---|
| SAMGPT | 47.76±10.55 | 55.57±12.04 | 56.33±9.04 | 73.36±4.52 |
| GFMate | **56.25±13.33** | **69.05±8.40** | **58.85±2.17** | **84.37±1.79** |
| △ | 8.49 | 13.48 | 2.52 | 10.01 |

## 5. Related Work

Our research domain generally intersects with three major areas: (i) **Graph Foundation Models (GFMs)**, which learn generalisable graph representations by label-free pre-training on large-scale source graphs, including LLM-based GFMs for text-attributed graphs (Liu et al., 2023a; Li et al., 2024; Chen et al., 2024a; Xia et al., 2024; Chen et al., 2024b; Kong et al., 2025) and GNN-based GFMs for general types of graphs without text attributes (Zhao et al., 2024a; Yu et al., 2024c; 2025; Sun et al., 2025; Wang et al., 2025). Since LLM-based GFMs are inherently limited to text-attributed graphs, this paper focuses on GNN-based GFMs due to their broader applicability. (ii) **Graph Prompt Tuning**, which introduces trainable prompts to align downstream tasks with pre-training objectives. Existing methods consist of single-domain graph prompts, which train prompts within a single domain, and cross-domain GFM prompts (Yu et al., 2024c; 2025; Wang et al., 2025), which learn prompts during pre-training on source domains and fine-tune them on unseen target domains. However, these prompts are usually entangled with source-domain information or specific pre-training strategies. (iii) **Test-time Methods for Graphs**, which aim to enhance the pre-trained model's performance on test graphs. Existing methods include test-time training (Sun et al., 2020; Wang et al., 2021; Liu et al., 2021; Zhang et al., 2022a;b; Chen et al., 2022; Wang et al., 2022; Zhang et al., 2024c;b; Zheng et al., 2024; Bao et al., 2024), which fine-tunes the model during inference, and test-time graph transformation (Jin et al., 2023; Ju et al., 2023), which modifies the test graphs. In contrast, GFMate performs test-time prompt tuning without modifying either the pre-trained GFM or the test graph. More details are in Appendix C.

## 6. Conclusion

This paper presents GFMate, a novel pre-training-agnostic test-time prompt tuning framework for GFMs. Motivated by empirical observation on existing methods, GFMate proposes centroid and layer prompts, enhancing prompt generalisability across domains and pre-trained GFMs. Moreover, a novel test-time complementary learning objective is introduced to exploit both labelled and unlabelled target domain data, mitigating the distribution shifts. Experiments on 12 benchmarks demonstrate its superior performance.

## Impact Statement

GFMate presents a novel pre-training-agnostic test-time graph prompt tuning framework that significantly enhances the cross-domain generalisability of Graph Foundation Models (GFMs). As a methodological advancement focused on graph learning efficiency, this work does not pose any ethical risks or negative societal impacts.

## Acknowledgments

This research has been supported by Australian Research Council Discovery Projects (DP230101196 and DE250100919).

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

# A. Supplementary Material Overview

In the Appendix, additional supplementary material to the main paper is provided. The structure is:

- The reproducibility statement is provided in Appendix B, including:
  - More detailed baseline method descriptions and categories in Table 7.
  - More detailed dataset descriptions and statistics in Table 5 and 6.
  - A Pseudo-code is provided in Algorithm 1.
  - More details on the implementation are included in the reproducibility statement.

- **More detailed related work is provided in Appendix C**, the structure is:
  - Appendix C.1 provides more detailed related work for pre-training methods on graphs.
  - Appendix C.2 provides more detailed related work for graph foundation models.
  - Appendix C.3 provides more detailed related work for prompting methods on graphs.
  - Appendix C.4 provides more detailed related work for test-time methods on graphs.
  - Appendix C.5 provides more detailed related work for test-time prompt tuning in other domains.

- Appendix D provides the theoretical analysis of the proposed test-time complementary learning.

- **More in-depth analysis of GFMate is provided in Appendix E**, the structure is:
  - Appendix E.1 provides the efficiency analysis for adaptation time, memory consumption, and parameter count.
  - Appendix E.2 evaluates the effectiveness of GFMate when less testing data are accessible.
  - Appendix E.3 provides the robustness analysis of GFMate under pre-training domain shift.
  - Appendix E.4 provides the robustness analysis of GFMate under noise and test-time distribution shift.
  - Appendix E.5 provides a t-SNE visualisation for the centroid with and without GFMate.
  - Appendix E.6 provides the sensitivity analysis of GFMate to hyperparameters.
  - Appendix E.7 provides the comparison between GFMate's test-time prompt tuning, conventional few-shot prompt tuning and pseudo label-based prompt tuning.
  - Appendix E.8 evaluates the accuracy of the test-time complementary labels by the layer-wise entropy-based strategies for the testing nodes.

- **More detailed experimental results are provided in Appendix F**, the structure is:
  - Appendix F.1 provides results for more baseline methods and results in 1, 3, 5, 10, and full-shot node classification scenarios.
  - Appendix F.2 provides results for more baseline methods on the graph classification task.

- A discussion of comparison between pre-training-entangled prompt and pre-training-agnostic prompt is provided in Appendix G.
- A discussion of the comparison between the existing few-shot prompt tuning and our test-time prompt tuning for GFMs is provided in Appendix H.
- A discussion of the detailed limitations is provided in Appendix I.

## B. Reproducibility Statement

To promote reproducible research, we summarise our efforts as follows:

- **Baselines.** We adopt baseline methods from existing GFM and prompt-based methods, including SAMGPT (Yu et al., 2025), DAGPrompt (Chen et al., 2025), and ProG benchmark (Zi et al., 2024), and carefully tune their hyperparameters via random search to optimise for a fair comparison. **All the existing methods are categorised and summarised in Table 7.**
- **Datasets.** We utilise 12 publicly available graph benchmark datasets from different domains, as in Table 5 and 6.
- **Algorithm.** Our GFMate framework is fully documented in the method section. In addition, we provide a detailed pseudo-code in Algorithm 1.
- **Implementation Details.** We utilise publicly available benchmark datasets, fully adhering to their CC-BY 4.0 license. The experiments are conducted using Python 3.8.2 and PyTorch 2.4.1 with CUDA 12.2, on a single NVIDIA A6000 GPU with 48GB of memory.

*Table 5.* **Datasets statistics for node classification.**

| Dataset | #Nodes | #Edges | #Feature | #Classes | #Full-shot | Domain | Source |
|---------|--------|--------|----------|----------|------------|--------|--------|
| Cornell | 183 | 554 | 1703 | 5 | 87 | Social Network | (Pei et al., 2020) |
| Texas | 183 | 558 | 1703 | 5 | 87 | Social Network | (Pei et al., 2020) |
| Wisconsin | 251 | 900 | 1703 | 5 | 120 | Social Network | (Pei et al., 2020) |
| Chameleon | 2277 | 36101 | 2325 | 5 | 1092 | Social Network | (Rozemberczki et al., 2021) |
| Squirrel | 5201 | 217073 | 2089 | 5 | 2496 | Social Network | (Rozemberczki et al., 2021) |
| Cora | 2708 | 10556 | 1433 | 7 | 140 | Citation Network | (Yang et al., 2016) |
| Citeseer | 3327 | 9104 | 3703 | 6 | 120 | Citation Network | (Yang et al., 2016) |
| Arxiv-year | 169343 | 1166243 | 128 | 5 | 100 | Citation Network | (Hu et al., 2020a) |
| Amazon-photo | 7650 | 238162 | 745 | 8 | 160 | Commercial System | (Shchur et al., 2018) |

*Table 6.* **Datasets statistics for graph classification.**

| Dataset | #Graphs | Avg.#Nodes | Avg.#Edges | #Feature | #Classes | Domain | Source |
|---------|---------|------------|------------|----------|----------|--------|--------|
| BZR | 405 | 35.8 | 38.4 | 3 | 2 | Biochemical Molecules | (Rossi & Ahmed, 2015) |
| COX2 | 467 | 41.2 | 43.5 | 3 | 2 | Biochemical Molecules | (Rossi & Ahmed, 2015) |
| PROTEINS | 1113 | 39.1 | 72.8 | 3 | 2 | Protein–Protein Interaction | (Borgwardt et al., 2005) |

---

**Algorithm 1: Pseudo-code of GFMate**

---

**Input:** Node features $\boldsymbol{X}$, adjacency $\boldsymbol{A}$;
Target domain node set $\mathcal{V}_{\text{Tar}}$ (including few-shot $\mathcal{V}_{\text{Fs}}$ and testing $\mathcal{V}_{\text{Te}}$);
Pre-trained GFM with fixed parameters $\boldsymbol{\theta}^*$;
Loss weight $\gamma$, learning rate $\alpha$
**Output:** Final predictions $\hat{\boldsymbol{y}}$ for testing nodes

1   % **Stage 1: Encoding Process**;
2   Encode nodes using fixed GFM to obtain embeddings $\boldsymbol{H}^{(l)}$ for $l \in \{0, \ldots, L\}$;
3   Initialize centroids $\boldsymbol{e}_c^{(l)}$ using few-shot mean (Eq. 3);

4   % **Stage 2: Pre-training-agnostic Prompt Design**;
5   Initialize centroid prompt $\boldsymbol{\beta}$ and layer prompt $\boldsymbol{\eta}$;
6   **for** $l \leftarrow 0$ **to** $L$ **do**
7     **for** $c \leftarrow 1$ **to** $C$ **do**
8       Refine centroid: $\widetilde{\boldsymbol{e}}_c^{(l)} \leftarrow \boldsymbol{e}_c^{(l)} + \boldsymbol{\beta}_c^{(l)}$ (Eq. 4);

9   % **Stage 3: Test-time Graph Complementary Learning**;
10   Compute layer-wise entropy $H^{(l)}$ to select pivot layer $\hat{l}$ (Eq. 7);
11   Construct complementary labels $\overline{y}$ based on $\hat{l}$;
12   **while** *not converged* **do**
13     Compute test-time learning loss $\mathcal{L}_{\text{Te}}$ on $\mathcal{V}_{\text{Te}}$ (Eq. 8);
14     Compute few-shot learning loss $\mathcal{L}_{\text{Fs}}$ on $\mathcal{V}_{\text{Fs}}$ (Eq. 9);
15     Calculate total objective $\mathcal{L}_{\text{TGCL}}$ (Eq. 10):;
16       $\mathcal{L}_{\text{TGCL}} \leftarrow \gamma \mathcal{L}_{\text{Te}} + (1 - \gamma)\mathcal{L}_{\text{Fs}}$;
17     Update prompts via gradient descent (Eq. 11):;
18       $\boldsymbol{\beta} \leftarrow \boldsymbol{\beta} - \alpha \nabla_{\boldsymbol{\beta}} \mathcal{L}_{\text{TGCL}}$;
19       $\boldsymbol{\eta} \leftarrow \boldsymbol{\eta} - \alpha \nabla_{\boldsymbol{\eta}} \mathcal{L}_{\text{TGCL}}$;

20   % **Stage 4: Prediction with Multi-layer Ensemble**;
21   **for** *each testing node* $i \in \mathcal{V}_{Te}$ **do**
22     Predict $\hat{y}_i \leftarrow \text{argmax}_c \left[ \text{Softmax} \left( \sum_{l=0}^{L} \eta^{(l)} \cdot \text{sim}(\boldsymbol{h}_i^{(l)}, \widetilde{\boldsymbol{E}}^{(l)}) \right) \right]$ (Eq. 6);

23   **return** $\hat{\boldsymbol{y}}$

---

*Table 7.* **Summary of existing methods for learning on graphs.**

| Generalisability | Categories | Representative Methods | Graphs |
|---|---|---|---|
| **Non-GFMs** | **Supervised GNNs** | GCN (Kipf & Welling, 2017), GAT (Veličković et al., 2018), SAGE (Hamilton et al., 2017) H2GCN (Zhu et al., 2020a), GPR (Chien et al., 2021) | **Text-free** |
| | **Supervised + Test-time** | GTrans (Jin et al., 2023), GraphPatcher (Ju et al., 2023) | |
| | **Linear GNNs** | GraphAny (Zhao et al., 2025) | |
| | **Pre-training + Fine-tuning** | Link Prediction (Lu et al., 2021), DGI (Velickovic et al., 2019), GCL (You et al., 2020) | |
| | **Pre-training + Prompt** | GPPT (Sun et al., 2022), All-In-One (Sun et al., 2023), ProNoG (Yu et al., 2024b) DAGPrompt (Chen et al., 2025), GraphPrompt (Liu et al., 2023b), GPF (Fang et al., 2024) | |
| **GFMs** | | GCOPE (Zhao et al., 2024a), MDGFM (Wang et al., 2025), MDGPT (Yu et al., 2024c) SAMGPT (Yu et al., 2025), BRIDGE (Yuan et al., 2025) | |
| | **Other GFMs** | RiemannGFM (Sun et al., 2025) | |
| | **GFMs via LLMs** | OFA (Liu et al., 2023a), ZeroG (Li et al., 2024), GraphWiz (Chen et al., 2024a) | **TAGs** |

# C. Detailed Related Work

## C.1. Graph Pre-training Methods

Graph pre-training methods seek to capture the intrinsic characteristics of graphs, predominantly leveraging self-supervised learning techniques. Such approaches can be categorised into two main types: (i) Graph Reconstruction Based Methods, which aim to recover specific graph attributes (Hu et al., 2020b; Hou et al., 2022; Kipf & Welling, 2016; Lu et al., 2021); (ii) Graph Contrastive Learning Based Methods, which improve representation learning by contrastive learning on different views of representations (Zhu et al., 2020b; Zeng & Xie, 2021; You et al., 2020; Velickovic et al., 2019; Sun et al., 2019). Despite their effectiveness, these approaches struggle to mitigate the task objective discrepancy between pre-training and fine-tuning, thus constraining their generalisation capacity on different downstream tasks (Sun et al., 2023; Zhao et al., 2024a).

## C.2. Graph Foundation Models

GFMs aim to learn generalisable graph representations by leveraging label-free pre-training on large-scale source graphs to adapt to a wide range of downstream tasks and target graphs in different domains (Liu et al., 2025). Current GFMs can be broadly categorised based on backbone architectures into (i) **LLM-based GFMs** (Liu et al., 2023a; Li et al., 2024; Chen et al., 2024a; Xia et al., 2024; Chen et al., 2024b; Kong et al., 2025) and (ii) **GNN-based GFMs** (Zhao et al., 2024a; Yu et al., 2024c; 2025; Sun et al., 2025; Wang et al., 2025; Yuan et al., 2025). LLM-based GFMs utilise textual information in graphs by harnessing the language modelling capabilities of large language models (LLMs) (Brown et al., 2020), but are inherently limited to text-attributed graphs (TAGs). In contrast, GNN-based GFMs operate in continuous feature spaces and are applicable to general text-free graphs (Zhao et al., 2024a; Sun et al., 2025; Wang et al., 2025; Yu et al., 2024c; 2025; Yuan et al., 2025). Therefore, GFMate focuses on enhancing the GNN-based GFMs due to their broader applicability.

## C.3. Graph Prompt Tuning Methods

**Conventional Single-domain Graph Prompt Tuning.** Graph prompting seeks to reformulate diverse downstream tasks to align with the pre-training objective by introducing trainable prompts, without requiring fine-tuning of the model (Sun et al., 2023). Existing graph prompt tuning approaches primarily target different types of graphs, such as homophilic graphs (Sun et al., 2022), heterophilic graphs (Yu et al., 2024b; Chen et al., 2025) and (ii) other types, including graphs from biological domains (Liu et al., 2023b; Yu et al., 2024a; Fang et al., 2024). Such prompting methods generally assume that source and target graphs originate from the same domain (Hu et al., 2020b; Qiu et al., 2020), thereby overlooking scenarios in which pre-training and downstream tasks involve datasets drawn from distinct domains.

**Cross-domain Graph Prompt Tuning for GFMs.** Recent advancements in graph prompt tuning have extended to the domain of GFMs, aiming to enhance their cross-domain generalisation by designing diverse prompting strategies (Wang et al., 2025; Yu et al., 2024c; 2025). MDGPT (Yu et al., 2024c) develops a unifying prompt and a mixing prompt to align the target domains with source domains. SAMGPT (Yu et al., 2025) further develops structural tokens with dual prompts during both pre-training and downstream stages to enhance the structural alignment between source and target domains. BRIDGE (Yuan et al., 2025) further proposes a domain invariant aligner during pre-training and employs a mixture of expert networks for cross-domain downstream prompting.

## C.4. Test-time Methods in Graph Domain

Test-time methods for graphs aim to enhance the pre-trained model's performance on test graphs. They can be broadly categorised into two types: (i) test-time training (Sun et al., 2020; Wang et al., 2021; Liu et al., 2021; Zhang et al., 2022a;b; Chen et al., 2022; Wang et al., 2022; Zhang et al., 2024c;b; Zheng et al., 2024; Bao et al., 2024), which adapt the model during inference without altering the input graph; and (ii) test-time graph transformation (Jin et al., 2023; Ju et al., 2023), which improve the test data by modifying the graph structure or node features without retraining the model. In contrast, GFMate diverges from both categories by performing test-time tuning of the additional prompts, without modifying either the model or the test graph, thereby introducing a novel and unexplored direction.

## C.5. Test-time Prompt Tuning in Other Domains

Test-time prompt tuning is a promising research direction to address distribution shifts during downstream tasks in domains such as computer vision (CV) (Shu et al., 2022; Ma et al., 2023; Hassan et al., 2024; Feng et al., 2023; Zhao et al., 2024b;

Zhang et al., 2024a). These methods leverage test data to optimise prompts, which are concatenated with input data and passed to the fixed model to enhance test-time performance. In comparison, GFMate optimises the graph prompts to enhance downstream task performance without altering the input data, which significantly differs from test-time prompt tuning methods in the CV domain.

# D. Detailed Proof for Theoretical Analysis

In this section, we prove the excess risk bound for the hypothesis of the proposed test-time complementary learning. Our test-time graph complementary learning objective falls into the category of training-time noisy label learning, since the ground-truth complementary labels on test data are inaccessible during test-time.

---

**Proposition 1 (Excess Risk Bound for Test-time Learning in GFMate.)**

Let $\mathcal{F}$ be the hypothesis space, where each predictor model $f \in \mathcal{F}$ refers to a GFM with prompts, defined as $f = (\text{GFM}_{\theta^*}, \mathcal{B})$ with pre-trained parameters $\theta^*$ and learnable prompts $\mathcal{B}$. Let $\overline{\mathcal{L}}$ be the test-time learning loss over test data $x$ and complementary labels $\overline{y}$. The population risk is defined as:

$$R_{\overline{\mathcal{L}}}(f) = \mathbb{E}_{(x,\overline{y})}\big[\overline{\mathcal{L}}(f(x), \overline{y})\big].$$

Define the empirical risk minimiser as $\hat{f} = \arg\min_{f \in \mathcal{F}} \hat{R}_{\overline{\mathcal{L}}}(f)$. With probability at least $1 - \delta$, the generalisation bound on the excess risk holds:

$$R_{\overline{\mathcal{L}}}(\hat{f}) - \min_{f \in \mathcal{F}} R_{\overline{\mathcal{L}}}(f) \leq 4C\ell_\rho \mathfrak{R}(\mathcal{F}) + 2\sqrt{\frac{\log(1/\delta)}{2N}}.$$

---

*Proof.* To simplify the notation, let $\mathcal{F}$ be the hypothesis space, where each $f \in \mathcal{F}$ denotes a predictor model of the form $f = (\text{GFM}_{\theta^*}, \mathcal{B})$ with fixed pre-trained model parameters $\theta^*$ and learnable prompt parameters $\mathcal{B}$. Let $\mathcal{L}(\hat{y}, y)$ be an $\ell$-Lipschitz loss function with respect to the predicted output $\hat{y}$ (for every label $y$). Then, with probability at least $1 - \delta$, the generalisation error based on the Rademacher complexity (Shalev-Shwartz & Ben-David, 2014) satisfies:

$$\sup_{f \in \mathcal{F}} \left| R_{\mathcal{L}}(f) - \hat{R}_{\mathcal{L}}(f) \right| \leq 2\ell_\rho \mathfrak{R}(\mathcal{F}) + \sqrt{\frac{\log(1/\delta)}{2N}}, \tag{11}$$

where $\mathfrak{R}(\mathcal{F}) := \mathbb{E}_{X_i, \epsilon_i}\big[\sup_{\text{GFM}_{\theta^*}, \mathcal{B} \in \mathcal{F}} \frac{1}{n} \sum \epsilon_i \text{GFM}_{\theta^*}, \mathcal{B}(X_i)\big]$ denotes the Rademacher complexity of $\mathcal{F}$, and $\ell_\rho \leq \frac{2\ell}{1 - \rho_{+1} - \rho_{-1}}$ is the Lipschitz constant of the loss $\mathcal{L}$ under label noise. Here, $\epsilon_i$ are i.i.d. Rademacher variables, and $n$ is the number of training samples.

Let $\hat{f} = \arg\min_{f \in \mathcal{F}} \hat{R}_{\mathcal{L}}(f)$ be the empirical risk minimiser, and let $f^* = \arg\min_{f \in \mathcal{F}} R_{\mathcal{L}}(f)$ denote the optimal hypothesis with respect to the true risk. Here, $R_{\mathcal{L}}(f)$ and $\hat{R}_{\mathcal{L}}(f)$ denote the true (expected) and empirical risks of a hypothesis $f$, respectively; $R_{\mathcal{L}}(\hat{f})$ and $\hat{R}_{\mathcal{L}}(\hat{f})$ are the true and empirical risks of the learned model $\hat{f}$; and $R_{\mathcal{L}}(f^*)$ and $\hat{R}_{\mathcal{L}}(f^*)$ denote the true and empirical risks of the optimal hypothesis $f^*$. Then, the following inequality holds:

$$\hat{R}_{\mathcal{L}}(\hat{f}) \leq \hat{R}_{\mathcal{L}}(f^*).$$

Because $\hat{R}_{\mathcal{L}}(\hat{f})$, defined as the empirical risk of $\hat{f}$, is the global minimiser of the empirical risk, it satisfies $\hat{R}_{\mathcal{L}}(\hat{f}) \leq \hat{R}_{\mathcal{L}}(f)$ for all $f \in \mathcal{F}$, including $f^*$. Therefore, the excess risk of $\hat{f}$ over $f^*$ can be bounded as:

$$\begin{aligned}
R_{\mathcal{L}}(\hat{f}) - R_{\mathcal{L}}(f^*) &= R_{\mathcal{L}}(\hat{f}) - \hat{R}_{\mathcal{L}}(\hat{f}) + \hat{R}_{\mathcal{L}}(\hat{f}) - R_{\mathcal{L}}(f^*) \\
&\leq R_{\mathcal{L}}(\hat{f}) - \hat{R}_{\mathcal{L}}(\hat{f}) + \hat{R}_{\mathcal{L}}(f^*) - R_{\mathcal{L}}(f^*) \\
&\leq 2 \sup_{f \in \mathcal{F}} \left| R_{\mathcal{L}}(f) - \hat{R}_{\mathcal{L}}(f) \right| \\
&\leq 4\ell_\rho \mathfrak{R}(\mathcal{F}) + 2\sqrt{\frac{\log(1/\delta)}{2N}}.
\end{aligned} \tag{12}$$

The above result is for the standard loss $\mathcal{L}$. To extend this to the complementary loss $\overline{\mathcal{L}}$, we note that our test-time loss aligns with the one-vs-all loss (Ishida et al., 2017), where the model is encouraged to distance itself from the complementary class while remaining close to others. Therefore, by Talagrand's contraction lemma (Ishida et al., 2017; 2019), the Rademacher complexity of the one-vs-all complementary loss $\overline{\mathcal{L}}$ is bounded by:

$$\overline{\mathfrak{R}}_n(\mathcal{F}_{\text{OVA}}) \leq C\ell_\rho \mathfrak{R}_n(\mathcal{F}),$$

where $C$ is the number of classes. Hence, with probability at least $1 - \delta$, the excess risk for test-time complementary learning is bounded as:

$$R_{\overline{\mathcal{L}}}(\hat{f}) - \min_{f \in \mathcal{F}} R_{\overline{\mathcal{L}}}(f) \leq 4C\ell_\rho \mathfrak{R}(\mathcal{F}) + 2\sqrt{\frac{\log(1/\delta)}{2N}}. \tag{13}$$

In summary, the excess risk of the test-time learning loss is upper bounded by a term that increases with the number of classes and decreases at a rate of $\mathcal{O}(1/\sqrt{N})$, where $N$ denotes the number of testing samples. Intuitively, Proposition 1 suggests that a smaller number of classes or a greater number of complementary-labelled samples leads to a tighter risk upper bound of the GFM prompt method. As GFMate utilises all unlabelled test data for learning, the bound becomes tighter, supporting the theoretical benefit of our test-time learning framework. $\qquad \square$

## E. Additional In-depth Analysis on GFMate

This section presents a more in-depth analysis of GFMate, including the more detailed efficiency analysis in Appendix E.1 and more detailed effectiveness analysis in Section 4.6.

*Table 8.* **Efficiency Analysis.** Downstream adaptation time in seconds for 5 repeat runs, GPU peak memory in MB and total number of tunable parameters. Average accuracy is also provided.

| | Cornell | | | | Citeseer | | | | Cora | | | |
|---|---|---|---|---|---|---|---|---|---|---|---|---|
| | Time (s) | Memory (MB) | #Params | Acc (%) | Time (s) | Memory (MB) | #Params | Acc (%) | Time (s) | Memory (MB) | #Params | Acc (%) |
| **GFMate** | 19 | 560 | 5124 | 79.67 | 153 | 1416 | 4611 | 55.27 | 269 | 1494 | 5379 | 59.68 |
| **Fine-tuning** | 53 | 646 | 1081350 | 23.88 | 109 | 994 | 1212934 | 33.39 | 102 | 936 | 1147142 | 29.85 |
| **DAGPrompt** | 140 | 876 | 31022 | 59.11 | 195 | 2100 | 31748 | 47.24 | 273 | 2096 | 32772 | 54.88 |
| **RiemannGFM** | 239 | 4364 | 109764 | 46.35 | 248 | 4599 | 123876 | 37.71 | 418 | 5721 | 146824 | 39.91 |
| **MDGPT** | 1275 | 18436 | 137645 | 54.19 | 1579 | 19108 | 147659 | 41.98 | 1494 | 20368 | 143742 | 44.52 |
| **SAMGPT** | 1076 | 19876 | 282360 | 59.34 | 1396 | 24599 | 283773 | 47.76 | 1267 | 13077 | 282929 | 52.83 |

### E.1. Detailed Efficiency Analysis

This section presents a detailed efficiency analysis to evaluate downstream adaptation time, peak GPU memory usage, and the number of tunable parameters, in relation to the average accuracy over five repeated runs on one-shot node classification tasks. To ensure a fair comparison, all time measurements start after the GFM model pre-training is completed, with the model fixed and the target domain graph becoming available. As shown in Table 8, GFMate demonstrates significantly faster downstream adaptation time and reduced GPU memory consumption. The number of tunable parameters in GFMate is notably lower than that in existing GFM methods like SAMGPT, prompting-based methods like DAGPrompt, and full fine-tuning approaches. This highlights the lightweight nature of the pre-training-agnostic prompts in GFMate compared to existing prompt-based methods. Overall, GFMate consistently achieves superior time and space efficiency, while delivering the highest accuracy across all baseline methods.

*Table 9.* **Effectiveness on GNN-based GFMs with different backbones under 3-shot setting.**

| | Texas | Cornell | Citeseer | Cora |
|---|---|---|---|---|
| **GFMate** | $83.29_{\pm1.52}$ | $88.57_{\pm0.87}$ | $74.08_{\pm1.72}$ | $70.51_{\pm2.11}$ |
| **w/ SAGE** | $82.76_{\pm1.21}$ | $88.24_{\pm1.26}$ | $73.63_{\pm1.89}$ | $69.35_{\pm2.94}$ |
| **w/ GAT** | $83.40_{\pm1.55}$ | $88.97_{\pm1.03}$ | $73.68_{\pm2.14}$ | $70.97_{\pm2.68}$ |
| **w/ H2GCN** | $85.56_{\pm2.29}$ | $90.07_{\pm1.58}$ | $70.74_{\pm3.25}$ | $65.82_{\pm3.47}$ |

### E.2. GFMate with less testing data

In this section, the effectiveness of GFMate when fewer unlabelled testing data are available is evaluated. According to Table 11, a larger number of testing nodes leads to better performance for GFMate. This arises from the effective test time

*Table 10.* **Plug-in GFMate on different pre-training methods under 3-shot setting.**

| | Texas | Cornell | Citeseer | Cora |
|---|---|---|---|---|
| **LP+FT** | $38.58_{\pm8.84}$ | $35.79_{\pm10.33}$ | $44.36_{\pm7.63}$ | $42.26_{\pm5.48}$ |
| **+GFMate** | $83.29_{\pm1.52}$ | $88.57_{\pm0.87}$ | $74.08_{\pm1.72}$ | $70.51_{\pm2.11}$ |
| **DGI+FT** | $40.33_{\pm7.81}$ | $34.48_{\pm9.97}$ | $41.77_{\pm7.90}$ | $39.98_{\pm6.33}$ |
| **+GFMate** | $81.38_{\pm3.94}$ | $88.19_{\pm4.04}$ | $73.70_{\pm5.42}$ | $62.25_{\pm3.52}$ |
| **GCL+FT** | $34.40_{\pm9.05}$ | $29.88_{\pm9.72}$ | $45.52_{\pm8.86}$ | $40.18_{\pm5.66}$ |
| **+GFMate** | $76.54_{\pm2.92}$ | $84.33_{\pm3.25}$ | $73.95_{\pm3.72}$ | $68.49_{\pm3.60}$ |
| **MDGPT** | $66.81_{\pm6.39}$ | $55.76_{\pm6.58}$ | $45.59_{\pm9.64}$ | $52.88_{\pm6.72}$ |
| **+GFMate** | $82.15_{\pm3.61}$ | $85.24_{\pm4.19}$ | $72.56_{\pm3.41}$ | $69.04_{\pm3.24}$ |
| **SAMGPT** | $70.44_{\pm7.61}$ | $72.25_{\pm7.67}$ | $49.58_{\pm8.71}$ | $63.39_{\pm7.71}$ |
| **+GFMate** | $80.60_{\pm2.03}$ | $85.72_{\pm3.21}$ | $73.30_{\pm4.52}$ | $70.10_{\pm4.63}$ |

complementary learning mechanism in GFMate, which successfully utilises the unlabelled target domain testing data to adapt the GFM. Moreover, this empirical observation aligns with the theoretical insight provided by Proposition 1, which indicates that a greater amount of testing data yields better generalisability for GFMate.

*Table 11.* **GFMate with less testing data consistently demonstrates its effectiveness.** The ratio indicates how many randomly selected testing nodes are available. GFMate at zero percent denotes the setting where GFMate uses few-shot tuning only.

| | Cornell | | Citeseer | | Cora | | Arxiv-year | |
|---|---|---|---|---|---|---|---|---|
| | 1-shot | 3-shot | 1-shot | 3-shot | 1-shot | 3-shot | 1-shot | 3-shot |
| **GFMate-0%** | $72.59_{\pm7.68}$ | $86.13_{\pm2.40}$ | $52.92_{\pm10.34}$ | $71.95_{\pm3.66}$ | $54.57_{\pm4.31}$ | $68.79_{\pm3.24}$ | $28.15_{\pm3.50}$ | $28.59_{\pm3.32}$ |
| **GFMate-20%** | $73.30_{\pm7.47}$ | $86.52_{\pm3.36}$ | $54.40_{\pm11.25}$ | $72.03_{\pm4.32}$ | $55.22_{\pm4.98}$ | $68.92_{\pm3.31}$ | $28.20_{\pm3.65}$ | $29.06_{\pm3.98}$ |
| **GFMate-50%** | $77.69_{\pm8.61}$ | $87.13_{\pm2.01}$ | $54.98_{\pm11.37}$ | $72.68_{\pm2.76}$ | $56.60_{\pm4.33}$ | $69.01_{\pm3.17}$ | $29.73_{\pm4.07}$ | $29.52_{\pm3.31}$ |
| **GFMate-80%** | $78.06_{\pm8.08}$ | $87.97_{\pm1.94}$ | $55.79_{\pm12.16}$ | $73.35_{\pm2.07}$ | $58.17_{\pm5.08}$ | $69.79_{\pm2.68}$ | $30.05_{\pm3.96}$ | $30.34_{\pm3.37}$ |
| **GFMate-Full** | $79.67_{\pm8.47}$ | $88.57_{\pm0.87}$ | $56.25_{\pm13.33}$ | $74.08_{\pm1.72}$ | $59.68_{\pm5.37}$ | $70.51_{\pm2.11}$ | $30.19_{\pm3.65}$ | $30.61_{\pm2.97}$ |

*Table 12.* Robustness to GFMs pre-training domain shift. The default GFM backbone is GCN, and the pre-training strategy is link prediction. Each category of domain refers to a set of pre-training graphs, as shown in Table 5, excluding the target domain graphs. Comme and Bioinfn refer to the graph in commercial and bio-information system domains.

| | Texas | | Cornell | | Citeseer | | Cora | |
|---|---|---|---|---|---|---|---|---|
| | 1-shot | 3-shot | 1-shot | 3-shot | 1-shot | 3-shot | 1-shot | 3-shot |
| **GFMate** | $76.63_{\pm7.81}$ | $83.29_{\pm1.52}$ | $79.67_{\pm8.47}$ | $88.57_{\pm0.87}$ | $56.25_{\pm13.33}$ | $74.08_{\pm1.72}$ | $59.68_{\pm5.37}$ | $70.51_{\pm2.11}$ |
| **w/ Social** | $77.42_{\pm7.59}$ | $83.10_{\pm1.58}$ | $79.31_{\pm8.29}$ | $87.95_{\pm1.14}$ | $52.33_{\pm13.18}$ | $69.89_{\pm2.25}$ | $57.64_{\pm6.31}$ | $68.80_{\pm2.55}$ |
| **w/ Citation** | $75.97_{\pm8.24}$ | $81.15_{\pm2.06}$ | $77.49_{\pm8.76}$ | $85.52_{\pm1.35}$ | $54.68_{\pm12.16}$ | $73.79_{\pm2.01}$ | $59.19_{\pm5.87}$ | $70.55_{\pm2.30}$ |
| **w/ Comme** | $76.09_{\pm7.66}$ | $82.16_{\pm2.03}$ | $74.85_{\pm9.12}$ | $83.47_{\pm2.79}$ | $52.94_{\pm10.63}$ | $70.28_{\pm3.39}$ | $56.98_{\pm7.02}$ | $67.19_{\pm3.14}$ |
| **w/ Bioinfo** | $73.22_{\pm9.01}$ | $79.98_{\pm3.27}$ | $77.45_{\pm9.28}$ | $83.59_{\pm2.95}$ | $48.09_{\pm13.32}$ | $66.84_{\pm4.17}$ | $55.10_{\pm8.23}$ | $65.85_{\pm3.98}$ |

### E.3. Robustness upon Domain Shifts

In GFMate, the learnable layer-wise prompts capture graph patterns from the target domain and adaptively adjust the contribution of each layer's prediction to the final ensemble output. According to Table 12, GFMate demonstrates effectiveness across GFMs pre-trained on different domain graphs, while models pre-trained on all domains consistently achieve strong performance. These results showcase that the learnable layer prompts can mitigate domain shift by effectively adapting GFMs pre-trained on different domains to the target domain graph.

*Table 13.* **Robustness upon different test noise ratios** (Left: Feature Shift, Right: Structural Shift).

| | Texas | Cornell | Citeseer | Cora | Texas | Cornell | Citeseer | Cora |
|---|---|---|---|---|---|---|---|---|
| **GFMate** | $76.63_{\pm7.81}$ | $79.67_{\pm8.47}$ | $56.25_{\pm13.33}$ | $59.68_{\pm5.37}$ | $76.63_{\pm7.81}$ | $79.67_{\pm8.47}$ | $56.25_{\pm13.33}$ | $59.68_{\pm5.37}$ |
| **w/ 10%** | $76.97_{\pm6.96}$ | $79.52_{\pm6.94}$ | $54.83_{\pm13.09}$ | $59.49_{\pm5.00}$ | $76.52_{\pm7.31}$ | $79.64_{\pm7.03}$ | $54.06_{\pm12.52}$ | $59.47_{\pm5.62}$ |
| **w/ 30%** | $72.58_{\pm9.68}$ | $79.10_{\pm7.91}$ | $54.48_{\pm12.50}$ | $59.29_{\pm5.59}$ | $75.96_{\pm8.24}$ | $79.62_{\pm7.15}$ | $53.82_{\pm12.30}$ | $59.06_{\pm6.13}$ |
| **w/ 50%** | $71.24_{\pm9.77}$ | $79.07_{\pm8.38}$ | $54.42_{\pm14.12}$ | $58.97_{\pm5.54}$ | $74.61_{\pm10.46}$ | $79.52_{\pm7.36}$ | $51.59_{\pm12.41}$ | $58.98_{\pm5.82}$ |

### E.4. Robustness upon Noise and Test-time Distribution Shift.

Experiments with both feature noise and structural perturbations at varying ratios on the target graphs are conducted to evaluate GFMate's capability in addressing distribution shifts at test time. Feature noise is introduced by randomly shuffling testing node features, and structural perturbations are introduced by randomly dropping edges connected to testing nodes, both disturbing the embeddings encoded by the fixed GFM. As shown in Table 13, even with a 50% perturbation rate, GFMate still maintains strong performance, especially on Cornell, Citeseer, and Cora, verifying its effectiveness in mitigating train–test distribution shift using both labelled and unlabelled target data.

### E.5. Visualisation of Centroids

A t-SNE visualisation is conducted to verify the effectiveness of GFMate in a one-shot node classification task on Cora. The GFM is fixed, and GFMate optimises the centroids via the proposed test-time prompt tuning modules without accessing

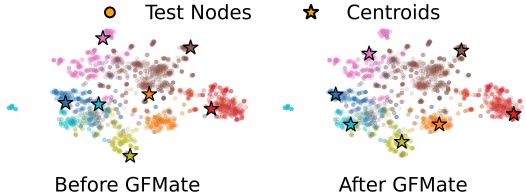

*Figure 8.* **t-SNE visualisation of centroids movement in GFMate.** After GFMate, the centroids become more closely aligned with the true class centres of the target-domain test nodes.

ground-truth test labels. As shown in Figure 8, after applying GFMate, the centroids align more closely with the true class centres of the testing nodes, contributing to improved classification performance.

### E.6. Hyperparameter Sensitivity Analysis

In this section, the impact of the hyperparameters $\gamma$ in GFMate is investigated. The parameter $\gamma$ controls the relative contribution of the test-time learning loss and the few-shot loss. The results of varying $\gamma$ in the one-shot node classification task are shown in Figure 9. Increasing $\gamma$ initially improves performance. However, further increases lead to a performance drop on Cora, while other datasets remain stable. This suggests that both the test-time and few-shot learning loss are essential, which aligns with the insight that learning from both labelled and unlabelled data is important.

### E.7. Comparison with Few-shot (Inductive Cases) and Pseudo Label Training

In this section, experiments are conducted to compare GFMate with conventional few-shot prompt tuning (using only the few-shot set to form a fully inductive setting where testing data is not accessible), as well as with the few-shot set augmented by test data with different ratios of pseudo labels, which are the most similar class from the last layer. According to Table 14, adopting pseudo labels for prompt tuning significantly degrades the performance of few-shot learning, owing to the inaccuracies in the pseudo labels. GFMate with test-time complementary learning substantially outperforms both few-shot prompt tuning and pseudo-label prompt tuning, validating the effectiveness of the proposed test-time graph complementary learning approach.

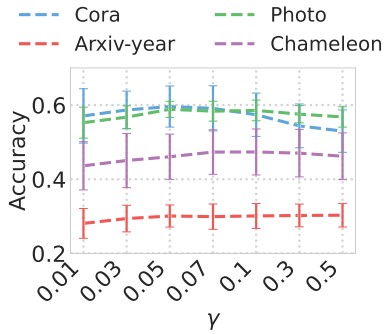

*Figure 9.* **Sensitivity studies of $\gamma$.**

It should be noted that when the test data are not accessible or in an inductive setting (which is not consistent with the transductive setting of all our baseline prompt methods (Yu et al., 2025; 2024c; Wang et al., 2025)), the test-time prompt tuning setting in GFMate reduces to a conventional few-shot prompt tuning scenario for the proposed pre-training agnostic prompts. Even in this extreme case, the GFMate variant tuned using only the few-shot set still outperforms the baseline method from Table 1, demonstrating that GFMate retains its effectiveness when test data are fully unavailable or in an inductive setting during GFM downstream adaptation.

*Table 14.* **Comparison of GFMate test-time prompt tuning with few-shot tuning (inductive setting) and pseudo-label tuning.** GFMate with test-time complementary learning is more effective than prompt tuning based on few-shot and pseudo labels of testing nodes. Few-shot denotes GFMate with prompt tuning only on the original few-shot set. Pseu denotes GFMate tuned with both few-shot and pseudo-labels of testing nodes from the last layer.

| | Cornell | | Citeseer | | Cora | | Arxiv-year | |
|---|---|---|---|---|---|---|---|---|
| | 1-shot | 3-shot | 1-shot | 3-shot | 1-shot | 3-shot | 1-shot | 3-shot |
| **GFMate** | $79.67_{\pm8.47}$ | $88.57_{\pm0.87}$ | $56.25_{\pm13.33}$ | $74.08_{\pm1.72}$ | $59.68_{\pm5.37}$ | $70.51_{\pm2.11}$ | $30.19_{\pm3.65}$ | $30.61_{\pm2.97}$ |
| **Few-shot** | $72.59_{\pm7.68}$ | $86.13_{\pm2.40}$ | $52.92_{\pm10.34}$ | $71.95_{\pm3.66}$ | $54.57_{\pm4.31}$ | $68.79_{\pm3.24}$ | $28.15_{\pm3.50}$ | $28.59_{\pm3.32}$ |
| **Pseu-All** | $57.95_{\pm9.36}$ | $58.05_{\pm4.27}$ | $13.79_{\pm8.91}$ | $52.91_{\pm6.43}$ | $18.29_{\pm12.35}$ | $35.97_{\pm10.64}$ | $12.16_{\pm8.04}$ | $15.61_{\pm8.63}$ |
| **Pseu-80%** | $58.10_{\pm9.27}$ | $58.39_{\pm3.91}$ | $22.58_{\pm7.72}$ | $59.50_{\pm4.28}$ | $25.53_{\pm9.81}$ | $39.49_{\pm8.28}$ | $13.77_{\pm8.40}$ | $16.48_{\pm7.25}$ |
| **Pseu-50%** | $60.37_{\pm8.96}$ | $62.33_{\pm3.65}$ | $35.84_{\pm7.65}$ | $63.27_{\pm5.04}$ | $38.39_{\pm9.47}$ | $44.71_{\pm7.66}$ | $18.52_{\pm6.58}$ | $20.29_{\pm7.50}$ |
| **Pseu-20%** | $66.95_{\pm8.81}$ | $67.72_{\pm4.31}$ | $45.27_{\pm9.89}$ | $69.96_{\pm4.37}$ | $46.61_{\pm7.15}$ | $59.75_{\pm9.54}$ | $24.07_{\pm5.39}$ | $24.32_{\pm7.11}$ |

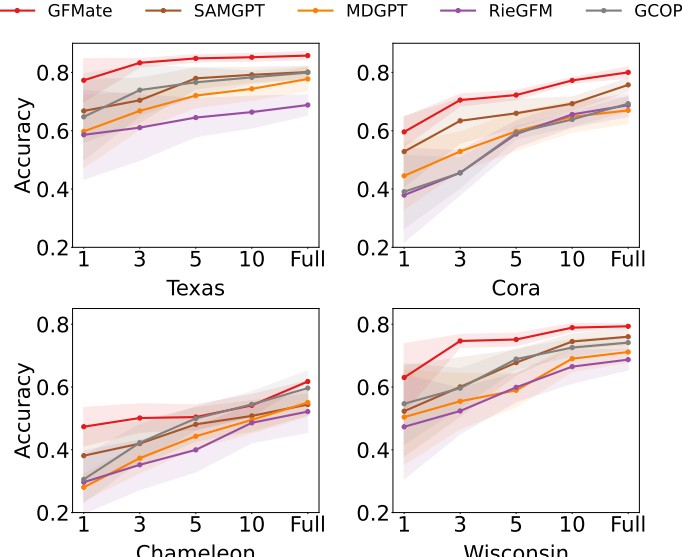

*Figure 10.* **Node classification performance of GFMate under different shot settings for more datasets.** RieGFM refers to RiemannGFM. GFMate consistently outperforms existing GFM methods across all scenarios.

### E.8. Test-time Complementary labels Analysis

In this section, the test-time complementary labels in GFMate are evaluated. GFMate assigns labels to testing nodes using a layer-wise entropy-based strategy, selecting the least similar class in the pivot layer $\hat{l}$ based on the entropy scores. According to Table 15, the complementary labels in GFMate are more accurate compared to those obtained by simply selecting the least similar class from the last layer, which showcases the effectiveness of the proposed layer-wise entropy-based strategies and contributes to the proposed test-time graph complementary learning.

*Table 15.* Accuracy of the test-time complementary label. GFMate samples test-time complementary labels using a layer-based entropy strategy, which is more accurate than simply selecting the least similar class as pseudo-label from the final layer.

| | **Cornell** | | **Citeseer** | | **Cora** | | **Arxiv-year** | |
| --- | --- | --- | --- | --- | --- | --- | --- | --- |
| | 1-shot | 3-shot | 1-shot | 3-shot | 1-shot | 3-shot | 1-shot | 3-shot |
| GFMate-comp | $99.08_{\pm0.27}$ | $99.57_{\pm0.15}$ | $99.25_{\pm0.49}$ | $99.36_{\pm0.31}$ | $96.63_{\pm0.86}$ | $97.20_{\pm0.63}$ | $88.14_{\pm1.29}$ | $90.26_{\pm1.14}$ |
| Pseudo-comp | $90.31_{\pm0.52}$ | $93.05_{\pm0.64}$ | $91.26_{\pm0.77}$ | $93.98_{\pm0.61}$ | $92.07_{\pm1.26}$ | $93.18_{\pm1.35}$ | $80.29_{\pm1.77}$ | $83.57_{\pm1.68}$ |

## F. Extended Experimental Results

This section provides additional experimental results, incorporating more baseline methods such as GTrans and GraphAny. To comprehensively evaluate the effectiveness of GFMate, the one-shot classification setting is extended to 3-shot, 5-shot, 10-shot, and full-shot settings. The full-shot setting corresponds to conventional supervised training, where the entire training set is utilised.

### F.1. Further Experimental Results on Node Classification

According to the detailed node classification results presented in Table 16, 17, 18, 19 and 20. "SL" refers to supervised learning-based GNN methods, while "TT" denotes test-time adaptation approaches. "Lin' indicates methods implemented using linear GNN architectures. "SSL+FT" represents self-supervised pre-training followed by supervised fine-tuning, whereas "SSL+Prompt" denotes self-supervised pre-training with prompt tuning. "GFMs" refer to methods designed for GFMs with cross-domain generalisability. GFMate demonstrates strong performance and consistently outperforms all baseline methods from 1-shot to full-shot settings, as demonstrated in Figure 10. Notably, when the target domain is the Cora or Amazon-Photo dataset, supervised training methods outperform all GFM and graph prompt tuning approaches, including GFMate. This observation aligns with prior findings that conventional supervised training becomes more effective

*Table 16.* **Full experimental results for node classification (Part 1).** GraphAny and GTrans are evaluated only under the full-shot setting, as they are not designed for GNNs trained in few-shot scenarios. GraphAny (C) and GraphAny (W) denote GraphAny trained on Cora and Wisconsin, respectively. Results are highlighted by **best** and runner-up.

| | Dataset | Texas | | | | | Cornell | | | | |
|---|---|---|---|---|---|---|---|---|---|---|---|
| | Setting | 1-shot | 3-shot | 5-shot | 10-shot | full-shot | 1-shot | 3-shot | 5-shot | 10-shot | full-shot |
| SL | GCN | 38.82±9.79 | 42.25±7.03 | 55.82±3.99 | 59.96±4.36 | 64.87±1.47 | 24.58±10.09 | 36.71±8.84 | 58.74±4.95 | 61.18±4.12 | 65.89±1.08 |
| | GAT | 39.96±9.62 | 41.19±6.64 | 55.72±2.68 | 59.17±3.32 | 65.41±1.08 | 25.84±12.26 | 37.11±10.61 | 60.88±4.99 | 65.53±4.10 | 68.92±5.51 |
| | SAGE | 40.38±8.21 | 44.39±8.11 | 68.81±3.55 | 70.25±2.24 | 81.32±3.59 | 32.55±13.36 | 40.81±8.89 | 70.41±5.33 | 80.04±3.99 | 82.83±3.15 |
| | GPR | 37.60±9.54 | 40.36±6.68 | 74.83±2.84 | 77.44±3.96 | 82.24±2.81 | 30.77±13.42 | 39.28±9.19 | 62.25±4.67 | 79.89±5.32 | 80.03±2.88 |
| | H2GCN | 47.75±12.89 | 56.69±7.83 | 70.66±3.59 | 76.99±4.51 | 80.65±3.69 | 35.58±15.02 | 42.29±9.60 | 69.90±5.53 | 80.33±5.57 | 84.40±4.37 |
| TT | GTrans | — | — | — | — | 67.79±4.33 | — | — | — | — | 65.72±5.84 |
| Lin | GraphAny (C) | — | — | — | — | 71.89±1.48 | — | — | — | — | 64.86±1.91 |
| | GraphAny (W) | — | — | — | — | 73.51±1.21 | — | — | — | — | 66.49±1.48 |
| SSL+FT | LP | 36.93±11.90 | 38.58±8.84 | 51.45±4.31 | 55.08±5.33 | 60.16±3.32 | 23.88±11.43 | 35.79±10.33 | 55.65±4.99 | 60.22±5.38 | 62.25±5.10 |
| | DGI | 34.46±12.92 | 40.33±7.81 | 50.73±4.39 | 54.46±4.98 | 59.85±3.34 | 22.89±11.32 | 34.48±9.97 | 52.26±6.60 | 61.32±5.34 | 63.35±4.89 |
| | GCL | 28.81±15.32 | 34.40±9.05 | 45.59±5.11 | 50.22±5.57 | 57.73±4.12 | 20.56±13.36 | 29.88±9.72 | 54.56±5.24 | 58.85±4.66 | 64.62±3.37 |
| SSL+Prompt | GPPT | 44.47±10.88 | 49.91±6.69 | 55.29±4.96 | 60.22±4.49 | 67.98±3.34 | 29.74±8.32 | 35.52±8.93 | 56.99±5.07 | 60.81±4.35 | 65.34±1.78 |
| | GPF | 47.34±12.72 | 55.87±8.62 | 61.49±4.33 | 65.02±4.17 | 68.86±3.58 | 52.15±14.53 | 59.07±11.48 | 69.92±7.74 | 74.08±7.90 | 82.25±5.88 |
| | GraphPrompt | 53.27±9.95 | 61.13±4.94 | 67.57±2.96 | 70.33±3.40 | 75.59±2.66 | 55.13±13.39 | 65.47±8.62 | 73.36±6.98 | 78.81±5.86 | 81.15±3.77 |
| | ProNoG | 55.31±12.92 | 57.99±8.48 | 72.24±5.98 | 77.36±3.77 | 80.36±4.43 | 48.49±11.54 | 69.91±7.62 | 79.94±6.68 | 82.25±5.03 | 85.59±3.31 |
| | DAGPrompt | 68.27±10.02 | 77.34±7.44 | 80.93±3.88 | 81.99±1.94 | 82.29±2.28 | 59.11±10.04 | 75.27±9.91 | 83.76±1.77 | 85.25±1.95 | 86.44±4.28 |
| | All-In-One | 63.79±15.91 | 69.44±9.19 | 74.92±5.79 | 75.22±2.54 | 78.11±2.25 | 57.24±12.70 | 65.35±6.48 | 75.50±6.33 | 80.64±5.44 | 82.29±3.90 |
| GFMs | GCOPE | 64.76±14.84 | 73.94±8.69 | 76.61±4.89 | 78.30±2.16 | 79.95±1.96 | 60.98±14.66 | 67.74±4.42 | 73.46±4.98 | 81.99±4.97 | 84.99±4.56 |
| | RiemannGFM | 58.60±15.27 | 61.06±11.24 | 64.51±6.58 | 66.39±5.47 | 68.82±3.49 | 46.35±11.92 | 52.20±5.79 | 58.93±6.14 | 69.97±6.08 | 78.36±5.21 |
| | MDGPT | 59.76±12.44 | 66.81±6.39 | 72.05±4.18 | 74.36±3.97 | 77.75±4.02 | 54.19±13.08 | 55.76±6.58 | 62.37±6.99 | 73.25±5.28 | 79.69±4.97 |
| | SAMGPT | 66.79±10.77 | 70.44±7.61 | 77.96±3.59 | 79.18±2.59 | 80.12±2.07 | 59.34±9.82 | 72.25±7.67 | 77.69±5.56 | 82.27±3.39 | 83.30±2.91 |
| | **GFMate** | **76.63±7.81** | **83.29±1.52** | **84.81±1.27** | **85.21±1.34** | **85.77±1.44** | **79.67±8.47** | **88.57±0.87** | **90.25±1.17** | **90.77±1.98** | **96.73±1.55** |

as more task-specific labelled data are available (Yu et al., 2025). In general, among GFM-based methods where the target domains differ from the source domains, GFMate achieves the best performance across all datasets and settings.

## F.2. Further Experimental Results on Graph Classification

In this section, the effectiveness of GFMate in the downstream graph classification task is evaluated. Methods such as GPPT, GTrans and GraphAny, which are not applicable to graph classification, are excluded from the comparison. According to Table 21, GFMate demonstrates the best performance compared to all baseline methods. GFMate also achieves substantially lower standard deviation, particularly on the COX2 and BZR datasets in biological target domains, where existing methods suffer from large standard deviations and consequently unstable performance. This indicates that GFMate's performance in graph classification is more stable than existing methods.

*Table 17.* **Full experimental results for node classification (Part 2).** GraphAny and GTrans are evaluated only under the full-shot setting, as they are not designed for GNNs trained in few-shot scenarios. GraphAny (C) and GraphAny (W) denote GraphAny trained on Cora and Wisconsin, respectively. Results are highlighted by **best** and runner-up.

| | Dataset | Chameleon | | | | | Wisconsin | | | | |
|---|---|---|---|---|---|---|---|---|---|---|---|
| | Setting | 1-shot | 3-shot | 5-shot | 10-shot | full-shot | 1-shot | 3-shot | 5-shot | 10-shot | full-shot |
| SL | GCN | $24.30_{\pm6.59}$ | $27.81_{\pm5.64}$ | $30.66_{\pm3.32}$ | $35.58_{\pm3.07}$ | $42.06_{\pm1.16}$ | $43.36_{\pm13.17}$ | $47.28_{\pm7.70}$ | $53.37_{\pm4.33}$ | $55.89_{\pm3.45}$ | $56.90_{\pm3.51}$ |
| | GAT | $22.81_{\pm7.38}$ | $28.14_{\pm4.46}$ | $31.17_{\pm4.09}$ | $37.72_{\pm3.30}$ | $44.34_{\pm1.55}$ | $45.99_{\pm10.86}$ | $49.91_{\pm6.18}$ | $51.49_{\pm4.08}$ | $52.63_{\pm4.43}$ | $54.51_{\pm3.37}$ |
| | SAGE | $31.29_{\pm8.87}$ | $38.21_{\pm4.37}$ | $45.82_{\pm2.26}$ | $49.89_{\pm2.95}$ | $51.54_{\pm0.55}$ | $47.41_{\pm11.36}$ | $49.24_{\pm8.61}$ | $58.88_{\pm4.05}$ | $69.66_{\pm3.34}$ | $83.14_{\pm1.56}$ |
| | GPR | $28.44_{\pm6.65}$ | $36.77_{\pm5.85}$ | $43.88_{\pm3.74}$ | $48.60_{\pm4.59}$ | $49.96_{\pm1.17}$ | $49.56_{\pm15.58}$ | $58.95_{\pm6.92}$ | $71.42_{\pm3.28}$ | $73.24_{\pm2.58}$ | $77.81_{\pm3.10}$ |
| | H2GCN | $28.01_{\pm9.50}$ | $33.75_{\pm5.88}$ | $44.89_{\pm6.01}$ | $47.31_{\pm3.38}$ | $52.01_{\pm2.65}$ | $45.55_{\pm10.61}$ | $55.33_{\pm6.94}$ | $65.16_{\pm3.35}$ | $73.36_{\pm3.88}$ | $76.69_{\pm4.13}$ |
| TT | GTrans | — | — | — | — | $41.82_{\pm2.29}$ | — | — | — | — | $59.98_{\pm4.96}$ |
| Lin | GraphAny (C) | — | — | — | — | $61.49_{\pm1.88}$ | — | — | — | — | $61.18_{\pm5.08}$ |
| | GraphAny (W) | — | — | — | — | $60.09_{\pm0.93}$ | — | — | — | — | $71.77_{\pm5.98}$ |
| SSL+FT | LP | $25.34_{\pm7.19}$ | $28.86_{\pm6.03}$ | $31.17_{\pm4.39}$ | $35.62_{\pm3.33}$ | $41.98_{\pm1.46}$ | $41.37_{\pm14.22}$ | $45.41_{\pm8.80}$ | $53.08_{\pm4.59}$ | $55.92_{\pm3.47}$ | $57.21_{\pm3.79}$ |
| | DGI | $25.78_{\pm7.34}$ | $29.04_{\pm6.11}$ | $31.59_{\pm4.46}$ | $37.02_{\pm4.18}$ | $42.34_{\pm2.80}$ | $38.89_{\pm15.77}$ | $42.31_{\pm10.52}$ | $51.19_{\pm5.57}$ | $54.96_{\pm3.55}$ | $55.73_{\pm3.68}$ |
| | GCL | $24.69_{\pm8.82}$ | $27.03_{\pm7.44}$ | $29.96_{\pm5.65}$ | $36.40_{\pm4.09}$ | $37.12_{\pm3.17}$ | $35.70_{\pm17.73}$ | $39.88_{\pm9.29}$ | $48.86_{\pm6.61}$ | $52.77_{\pm5.10}$ | $53.19_{\pm4.33}$ |
| SSL+Prompt | GPPT | $29.91_{\pm6.48}$ | $34.49_{\pm3.72}$ | $38.04_{\pm2.57}$ | $42.29_{\pm2.55}$ | $47.58_{\pm2.74}$ | $35.16_{\pm9.07}$ | $45.82_{\pm8.89}$ | $53.09_{\pm4.53}$ | $54.88_{\pm3.90}$ | $55.72_{\pm4.07}$ |
| | GPF | $30.95_{\pm9.18}$ | $36.76_{\pm5.33}$ | $45.44_{\pm2.37}$ | $47.32_{\pm2.86}$ | $49.90_{\pm2.34}$ | $40.19_{\pm14.49}$ | $47.22_{\pm9.34}$ | $58.86_{\pm4.41}$ | $60.33_{\pm3.78}$ | $68.84_{\pm4.02}$ |
| | GraphPrompt | $33.29_{\pm9.19}$ | $42.13_{\pm6.07}$ | $47.81_{\pm4.11}$ | $51.96_{\pm3.09}$ | $53.84_{\pm2.69}$ | $45.03_{\pm11.83}$ | $52.11_{\pm6.24}$ | $65.08_{\pm3.17}$ | $68.80_{\pm2.93}$ | $72.55_{\pm2.45}$ |
| | ProNoG | $31.19_{\pm8.09}$ | $35.52_{\pm6.63}$ | $37.34_{\pm3.88}$ | $38.79_{\pm2.21}$ | $43.38_{\pm2.95}$ | $46.29_{\pm17.74}$ | $55.84_{\pm8.81}$ | $66.51_{\pm4.33}$ | $70.22_{\pm3.70}$ | $75.97_{\pm2.26}$ |
| | DAGPrompt | $37.79_{\pm6.62}$ | $41.26_{\pm5.89}$ | $49.96_{\pm1.98}$ | $55.32_{\pm2.39}$ | $60.79_{\pm3.35}$ | $50.49_{\pm11.59}$ | $72.21_{\pm3.17}$ | $74.08_{\pm3.06}$ | $76.22_{\pm2.34}$ | $77.29_{\pm1.89}$ |
| | All-In-One | $27.94_{\pm6.31}$ | $38.03_{\pm4.75}$ | $46.33_{\pm3.05}$ | $52.20_{\pm2.14}$ | $58.85_{\pm1.17}$ | $55.35_{\pm18.36}$ | $67.17_{\pm4.28}$ | $69.22_{\pm3.75}$ | $70.40_{\pm2.29}$ | $72.01_{\pm1.68}$ |
| GFMs | GCOPE | $30.58_{\pm7.44}$ | $42.25_{\pm5.82}$ | $49.98_{\pm3.66}$ | $54.46_{\pm4.09}$ | $59.66_{\pm5.52}$ | $54.66_{\pm12.86}$ | $59.64_{\pm6.20}$ | $68.89_{\pm3.25}$ | $72.56_{\pm2.66}$ | $74.14_{\pm1.99}$ |
| | RiemannGFM | $29.68_{\pm9.95}$ | $35.21_{\pm7.88}$ | $39.97_{\pm7.04}$ | $48.59_{\pm6.34}$ | $52.17_{\pm6.65}$ | $47.32_{\pm16.58}$ | $52.36_{\pm7.28}$ | $59.90_{\pm4.56}$ | $66.49_{\pm5.32}$ | $68.72_{\pm3.31}$ |
| | MDGPT | $28.04_{\pm4.28}$ | $37.34_{\pm4.71}$ | $44.31_{\pm3.82}$ | $49.52_{\pm4.28}$ | $55.06_{\pm4.35}$ | $50.40_{\pm15.07}$ | $55.46_{\pm8.79}$ | $58.95_{\pm5.30}$ | $69.02_{\pm4.18}$ | $71.15_{\pm3.27}$ |
| | SAMGPT | $38.12_{\pm8.90}$ | $41.98_{\pm7.04}$ | $48.11_{\pm5.34}$ | $50.79_{\pm4.88}$ | $54.43_{\pm3.37}$ | $52.29_{\pm14.40}$ | $60.03_{\pm9.36}$ | $67.78_{\pm4.03}$ | $74.48_{\pm2.40}$ | $76.02_{\pm2.45}$ |
| | **GFMate** | $\mathbf{47.25_{\pm6.11}}$ | $\mathbf{50.12_{\pm4.55}}$ | $\mathbf{50.42_{\pm3.96}}$ | $\mathbf{54.15_{\pm3.49}}$ | $\mathbf{61.75_{\pm0.24}}$ | $\mathbf{63.01_{\pm10.78}}$ | $\mathbf{74.66_{\pm2.07}}$ | $\mathbf{75.13_{\pm2.03}}$ | $\mathbf{78.91_{\pm1.32}}$ | $\mathbf{79.34_{\pm0.77}}$ |

*Table 18.* **Full experimental results for node classification (Part 3).** GraphAny and GTrans are evaluated only under the full-shot setting, as they are not designed for GNNs trained in few-shot scenarios. GraphAny (C) and GraphAny (W) denote GraphAny trained on Cora and Wisconsin, respectively. Results are highlighted by **best** and runner-up.

| | Dataset | Squirrel | | | | | Amazon-photo | | | | |
|---|---|---|---|---|---|---|---|---|---|---|---|
| | Setting | 1-shot | 3-shot | 5-shot | 10-shot | full-shot | 1-shot | 3-shot | 5-shot | 10-shot | full-shot |
| SL | GCN | $19.96_{\pm5.89}$ | $20.14_{\pm2.37}$ | $22.08_{\pm2.22}$ | $24.49_{\pm1.08}$ | $28.88_{\pm1.15}$ | $47.09_{\pm5.81}$ | $58.38_{\pm4.08}$ | $61.09_{\pm4.34}$ | $73.35_{\pm3.39}$ | $89.75_{\pm0.77}$ |
| | GAT | $20.77_{\pm4.48}$ | $22.69_{\pm4.70}$ | $23.26_{\pm3.45}$ | $25.57_{\pm2.33}$ | $31.70_{\pm1.30}$ | $47.33_{\pm4.97}$ | $58.49_{\pm4.65}$ | $62.25_{\pm3.52}$ | $72.65_{\pm2.33}$ | $88.97_{\pm1.98}$ |
| | SAGE | $22.34_{\pm4.32}$ | $24.56_{\pm3.28}$ | $28.99_{\pm2.94}$ | $32.25_{\pm1.32}$ | $38.17_{\pm0.77}$ | $49.72_{\pm3.81}$ | $60.33_{\pm3.73}$ | $64.58_{\pm3.35}$ | $73.99_{\pm1.45}$ | **$90.61_{\pm0.59}$** |
| | GPR | $21.06_{\pm6.14}$ | $23.31_{\pm1.89}$ | $27.50_{\pm0.94}$ | $31.68_{\pm1.03}$ | $32.99_{\pm0.87}$ | $43.39_{\pm4.66}$ | $52.66_{\pm3.44}$ | $59.91_{\pm3.47}$ | $72.89_{\pm3.60}$ | $85.39_{\pm1.57}$ |
| | H2GCN | $21.10_{\pm3.06}$ | $25.57_{\pm4.47}$ | $27.95_{\pm4.06}$ | $29.91_{\pm3.44}$ | $35.25_{\pm2.40}$ | $45.81_{\pm6.09}$ | $58.86_{\pm5.33}$ | $66.74_{\pm4.88}$ | $72.26_{\pm2.84}$ | $89.05_{\pm2.26}$ |
| TT | GTrans | — | — | — | — | $30.70_{\pm2.51}$ | — | — | — | — | $90.06_{\pm1.44}$ |
| Lin | GraphAny (C) | — | — | — | — | $48.49_{\pm0.98}$ | — | — | — | — | $90.14_{\pm0.93}$ |
| | GraphAny (W) | — | — | — | — | $42.34_{\pm3.46}$ | — | — | — | — | $90.18_{\pm0.91}$ |
| SSL+FT | LP | $20.04_{\pm5.61}$ | $21.11_{\pm3.69}$ | $22.48_{\pm3.35}$ | $25.07_{\pm2.09}$ | $28.64_{\pm1.98}$ | $48.82_{\pm6.55}$ | $56.88_{\pm5.36}$ | $61.18_{\pm4.50}$ | $73.72_{\pm3.31}$ | $87.64_{\pm0.96}$ |
| | DGI | $20.85_{\pm6.57}$ | $21.28_{\pm4.06}$ | $22.35_{\pm3.39}$ | $24.98_{\pm2.44}$ | $27.65_{\pm2.03}$ | $45.17_{\pm7.34}$ | $56.67_{\pm6.12}$ | $60.09_{\pm3.89}$ | $72.54_{\pm3.59}$ | $85.25_{\pm1.34}$ |
| | GCL | $19.97_{\pm5.62}$ | $20.88_{\pm3.39}$ | $21.58_{\pm3.32}$ | $23.34_{\pm2.60}$ | $25.53_{\pm2.71}$ | $47.33_{\pm5.89}$ | $58.34_{\pm4.91}$ | $61.16_{\pm4.43}$ | $74.08_{\pm4.47}$ | $89.07_{\pm3.43}$ |
| SSL+Prompt | GPPT | $21.16_{\pm5.95}$ | $22.59_{\pm2.87}$ | $24.93_{\pm2.38}$ | $27.94_{\pm2.45}$ | $29.08_{\pm2.70}$ | $50.19_{\pm7.74}$ | $54.65_{\pm6.24}$ | $59.87_{\pm5.08}$ | $73.40_{\pm3.73}$ | $69.98_{\pm3.36}$ |
| | GPF | $22.71_{\pm4.87}$ | $25.26_{\pm3.99}$ | $27.95_{\pm2.80}$ | $29.94_{\pm3.31}$ | $30.49_{\pm2.79}$ | $49.38_{\pm6.56}$ | $53.37_{\pm6.52}$ | $62.29_{\pm7.03}$ | $73.88_{\pm5.82}$ | $66.70_{\pm4.56}$ |
| | GraphPrompt | $23.02_{\pm4.89}$ | $26.10_{\pm5.17}$ | $30.08_{\pm3.43}$ | $32.28_{\pm2.99}$ | $34.40_{\pm2.56}$ | $46.65_{\pm6.53}$ | $55.62_{\pm7.08}$ | $60.74_{\pm6.50}$ | $67.99_{\pm4.78}$ | $64.40_{\pm4.34}$ |
| | ProNoG | $24.25_{\pm4.79}$ | $27.97_{\pm4.40}$ | $28.68_{\pm3.49}$ | $30.04_{\pm2.99}$ | $32.25_{\pm2.33}$ | $47.72_{\pm6.60}$ | $52.85_{\pm6.68}$ | $60.69_{\pm5.88}$ | $63.98_{\pm6.63}$ | $69.92_{\pm2.79}$ |
| | DAGPrompt | $25.67_{\pm6.34}$ | $28.79_{\pm4.01}$ | $30.99_{\pm3.27}$ | $34.62_{\pm2.27}$ | $37.33_{\pm3.34}$ | $52.96_{\pm6.07}$ | $60.72_{\pm7.14}$ | $65.77_{\pm5.28}$ | $66.04_{\pm4.99}$ | $71.66_{\pm3.98}$ |
| | All-In-One | $21.18_{\pm7.06}$ | $24.32_{\pm5.03}$ | $29.58_{\pm4.88}$ | $31.45_{\pm3.04}$ | $35.59_{\pm3.92}$ | $52.25_{\pm7.33}$ | $54.84_{\pm6.36}$ | $63.67_{\pm5.52}$ | $66.32_{\pm5.65}$ | $67.75_{\pm2.44}$ |
| GFMs | GCOPE | $22.16_{\pm5.77}$ | $23.25_{\pm5.65}$ | $30.79_{\pm4.43}$ | $34.45_{\pm2.88}$ | $36.01_{\pm2.12}$ | $55.69_{\pm4.68}$ | $61.99_{\pm4.11}$ | $64.29_{\pm3.77}$ | $70.79_{\pm4.31}$ | $72.33_{\pm2.20}$ |
| | RiemannGFM | $20.13_{\pm8.58}$ | $22.08_{\pm7.35}$ | $25.39_{\pm5.52}$ | $27.30_{\pm5.08}$ | $29.79_{\pm5.27}$ | $49.69_{\pm13.32}$ | $52.18_{\pm9.44}$ | $58.47_{\pm6.36}$ | $64.15_{\pm6.08}$ | $68.99_{\pm5.61}$ |
| | MDGPT | $24.41_{\pm7.01}$ | $26.62_{\pm6.94}$ | $28.85_{\pm5.69}$ | $32.15_{\pm4.57}$ | $37.30_{\pm4.45}$ | $54.96_{\pm10.25}$ | $58.99_{\pm10.28}$ | $61.59_{\pm7.68}$ | $65.52_{\pm6.33}$ | $69.12_{\pm5.07}$ |
| | SAMGPT | $25.75_{\pm6.29}$ | $29.02_{\pm4.96}$ | $31.17_{\pm4.03}$ | $34.49_{\pm3.04}$ | $39.98_{\pm4.35}$ | $56.33_{\pm9.04}$ | $63.95_{\pm8.89}$ | $67.27_{\pm6.82}$ | $73.95_{\pm5.65}$ | $77.82_{\pm4.49}$ |
| | **GFMate** | **$27.02_{\pm6.22}$** | **$30.99_{\pm4.75}$** | **$32.77_{\pm3.58}$** | **$38.73_{\pm2.06}$** | **$42.44_{\pm1.73}$** | **$58.85_{\pm2.17}$** | **$64.28_{\pm4.72}$** | **$67.49_{\pm2.97}$** | **$74.40_{\pm0.95}$** | $78.06_{\pm2.69}$ |

*Table 19.* **Full experimental results for node classification (Part 4).** GraphAny and GTrans are evaluated only under the full-shot setting, as they are not designed for GNNs trained in few-shot scenarios. GraphAny (C) and GraphAny (W) denote GraphAny trained on Cora and Wisconsin, respectively. Results are highlighted by **best** and runner-up.

| | Dataset | Cora | | | | | Citeseer | | | | |
|---|---|---|---|---|---|---|---|---|---|---|---|
| | Setting | 1-shot | 3-shot | 5-shot | 10-shot | full-shot | 1-shot | 3-shot | 5-shot | 10-shot | full-shot |
| **SL** | GCN | $29.85_{\pm 8.98}$ | $35.26_{\pm 4.77}$ | $50.48_{\pm 3.39}$ | $58.79_{\pm 2.14}$ | $81.09_{\pm 1.18}$ | $33.39_{\pm 11.86}$ | $43.66_{\pm 6.80}$ | $45.71_{\pm 5.52}$ | $53.39_{\pm 2.10}$ | $68.18_{\pm 0.97}$ |
| | GAT | $33.25_{\pm 9.72}$ | $49.92_{\pm 5.33}$ | $55.78_{\pm 4.03}$ | $68.85_{\pm 3.63}$ | $80.47_{\pm 1.22}$ | $35.51_{\pm 9.70}$ | $40.11_{\pm 4.78}$ | $45.59_{\pm 3.89}$ | $57.12_{\pm 1.96}$ | $67.06_{\pm 2.19}$ |
| | SAGE | $35.76_{\pm 8.89}$ | $50.34_{\pm 4.88}$ | $65.41_{\pm 3.92}$ | $72.25_{\pm 2.58}$ | **$81.98_{\pm 1.56}$** | $38.80_{\pm 9.73}$ | $42.15_{\pm 3.56}$ | $48.81_{\pm 1.66}$ | $59.99_{\pm 1.43}$ | $68.69_{\pm 0.79}$ |
| | GPR | $38.99_{\pm 15.77}$ | $50.49_{\pm 9.49}$ | $64.25_{\pm 2.44}$ | $70.36_{\pm 1.85}$ | $79.43_{\pm 2.06}$ | $29.77_{\pm 13.22}$ | $33.89_{\pm 7.75}$ | $39.79_{\pm 4.88}$ | $48.97_{\pm 3.25}$ | $61.33_{\pm 1.03}$ |
| | H2GCN | $30.90_{\pm 9.98}$ | $48.86_{\pm 5.05}$ | $57.85_{\pm 3.44}$ | $69.77_{\pm 2.99}$ | $80.79_{\pm 1.35}$ | $30.91_{\pm 12.79}$ | $39.43_{\pm 8.02}$ | $45.58_{\pm 4.37}$ | $58.82_{\pm 3.22}$ | $67.42_{\pm 1.75}$ |
| **TT** | GTrans | — | — | — | — | $80.79_{\pm 2.43}$ | — | — | — | — | $69.94_{\pm 1.89}$ |
| **Lin** | GraphAny (C) | — | — | — | — | $79.98_{\pm 0.36}$ | — | — | — | — | $68.90_{\pm 0.07}$ |
| | GraphAny (W) | — | — | — | — | $77.82_{\pm 1.15}$ | — | — | — | — | $67.50_{\pm 0.44}$ |
| **SSL+FT** | LP | $35.59_{\pm 9.74}$ | $42.26_{\pm 5.48}$ | $59.98_{\pm 3.79}$ | $64.30_{\pm 2.84}$ | $79.38_{\pm 2.75}$ | $34.92_{\pm 12.08}$ | $44.36_{\pm 7.63}$ | $49.79_{\pm 5.65}$ | $57.64_{\pm 3.27}$ | $68.78_{\pm 2.15}$ |
| | DGI | $32.38_{\pm 8.86}$ | $39.98_{\pm 6.33}$ | $57.34_{\pm 2.81}$ | $64.46_{\pm 2.12}$ | $77.26_{\pm 2.59}$ | $33.96_{\pm 11.57}$ | $41.77_{\pm 7.90}$ | $47.62_{\pm 4.89}$ | $55.43_{\pm 3.31}$ | $65.46_{\pm 1.92}$ |
| | GCL | $33.27_{\pm 9.50}$ | $40.18_{\pm 5.66}$ | $55.37_{\pm 3.32}$ | $62.65_{\pm 2.58}$ | $75.58_{\pm 3.36}$ | $36.05_{\pm 13.44}$ | $45.52_{\pm 8.86}$ | $50.49_{\pm 5.61}$ | $58.84_{\pm 4.28}$ | $66.35_{\pm 2.44}$ |
| **SSL+Prompt** | GPPT | $40.62_{\pm 8.69}$ | $49.40_{\pm 4.71}$ | $63.39_{\pm 2.02}$ | $69.84_{\pm 1.92}$ | $71.19_{\pm 1.78}$ | $39.79_{\pm 10.67}$ | $47.65_{\pm 6.83}$ | $55.48_{\pm 5.16}$ | $59.89_{\pm 3.55}$ | $64.49_{\pm 3.13}$ |
| | GPF | $45.75_{\pm 9.61}$ | $52.29_{\pm 5.48}$ | $65.15_{\pm 2.58}$ | $67.35_{\pm 3.03}$ | $72.16_{\pm 2.28}$ | $40.51_{\pm 12.79}$ | $49.07_{\pm 7.62}$ | $57.35_{\pm 5.68}$ | $59.88_{\pm 4.55}$ | $62.75_{\pm 3.06}$ |
| | GraphPrompt | $49.77_{\pm 8.82}$ | $58.84_{\pm 4.98}$ | $70.34_{\pm 2.69}$ | $71.76_{\pm 2.27}$ | $74.51_{\pm 1.99}$ | $38.69_{\pm 13.98}$ | $45.16_{\pm 6.67}$ | $49.95_{\pm 6.60}$ | $55.83_{\pm 5.44}$ | $59.98_{\pm 5.14}$ |
| | ProNoG | $56.54_{\pm 12.33}$ | $59.87_{\pm 8.84}$ | $68.83_{\pm 3.40}$ | $71.35_{\pm 1.98}$ | $74.41_{\pm 2.25}$ | $37.79_{\pm 13.35}$ | $40.06_{\pm 7.14}$ | $53.39_{\pm 7.33}$ | $60.96_{\pm 4.99}$ | $62.29_{\pm 5.40}$ |
| | DAGPrompt | $54.88_{\pm 9.24}$ | $62.59_{\pm 5.78}$ | $70.19_{\pm 2.08}$ | $72.98_{\pm 0.99}$ | $75.08_{\pm 1.13}$ | $47.24_{\pm 9.59}$ | $60.99_{\pm 6.82}$ | $62.57_{\pm 4.96}$ | $66.72_{\pm 2.77}$ | $68.16_{\pm 3.25}$ |
| | All-In-One | $49.92_{\pm 11.75}$ | $50.44_{\pm 4.39}$ | $66.43_{\pm 2.21}$ | $69.51_{\pm 1.17}$ | $70.34_{\pm 1.42}$ | $40.69_{\pm 15.88}$ | $48.05_{\pm 6.20}$ | $53.25_{\pm 5.68}$ | $62.23_{\pm 4.59}$ | $64.49_{\pm 3.34}$ |
| **GFMs** | GCOPE | $39.06_{\pm 12.52}$ | $45.51_{\pm 5.17}$ | $59.32_{\pm 2.16}$ | $63.86_{\pm 2.39}$ | $69.28_{\pm 2.50}$ | $42.26_{\pm 14.19}$ | $50.35_{\pm 7.79}$ | $57.91_{\pm 6.77}$ | $65.70_{\pm 4.02}$ | $67.72_{\pm 2.25}$ |
| | RiemannGFM | $37.91_{\pm 16.13}$ | $45.58_{\pm 7.60}$ | $58.85_{\pm 4.52}$ | $65.59_{\pm 4.39}$ | $68.75_{\pm 3.84}$ | $38.02_{\pm 9.58}$ | $42.19_{\pm 7.67}$ | $54.03_{\pm 4.79}$ | $59.98_{\pm 3.72}$ | $61.46_{\pm 3.41}$ |
| | MDGPT | $44.52_{\pm 11.39}$ | $52.88_{\pm 6.72}$ | $59.79_{\pm 6.55}$ | $64.77_{\pm 5.28}$ | $66.98_{\pm 4.55}$ | $41.98_{\pm 12.24}$ | $45.59_{\pm 9.64}$ | $49.82_{\pm 6.33}$ | $55.79_{\pm 6.17}$ | $62.88_{\pm 4.82}$ |
| | SAMGPT | $52.83_{\pm 12.04}$ | $63.39_{\pm 7.71}$ | $65.98_{\pm 4.83}$ | $69.27_{\pm 2.31}$ | $75.72_{\pm 2.24}$ | $47.76_{\pm 13.06}$ | $49.58_{\pm 8.71}$ | $52.25_{\pm 5.58}$ | $61.17_{\pm 3.29}$ | $65.87_{\pm 1.98}$ |
| | **GFMate** | **$59.68_{\pm 5.37}$** | **$70.51_{\pm 2.11}$** | **$72.23_{\pm 1.61}$** | **$77.25_{\pm 1.03}$** | $80.02_{\pm 1.97}$ | **$56.25_{\pm 13.33}$** | **$74.08_{\pm 1.72}$** | **$76.38_{\pm 0.52}$** | **$77.23_{\pm 0.88}$** | **$78.34_{\pm 1.07}$** |

*Table 20.* **Full experimental results for node classification (Part 5) on large-scale dataset.** GraphAny and GTrans are evaluated only under the full-shot setting, as they are not designed for GNNs trained in few-shot scenarios. GraphAny (C) and GraphAny (W) denote GraphAny trained on Cora and Wisconsin, respectively. Results are highlighted by **best** and runner-up.

| | Dataset | Arxiv-year | | | | |
|---|---|---|---|---|---|---|
| | Setting | 1-shot | 3-shot | 5-shot | 10-shot | full-shot |
| SL | GCN | $18.64_{\pm6.91}$ | $20.26_{\pm4.74}$ | $23.17_{\pm2.11}$ | $27.39_{\pm1.96}$ | $33.44_{\pm1.98}$ |
| | GAT | $19.93_{\pm5.40}$ | $21.17_{\pm3.79}$ | $24.31_{\pm3.19}$ | $25.59_{\pm4.33}$ | $32.84_{\pm2.15}$ |
| | SAGE | $20.75_{\pm4.99}$ | $24.52_{\pm5.17}$ | $25.08_{\pm3.24}$ | $29.98_{\pm3.66}$ | $34.30_{\pm3.06}$ |
| | GPR | $10.81_{\pm9.94}$ | $15.45_{\pm5.57}$ | $20.37_{\pm4.51}$ | $24.46_{\pm3.39}$ | $28.77_{\pm2.58}$ |
| | H2GCN | $15.54_{\pm7.33}$ | $17.89_{\pm6.48}$ | $22.35_{\pm3.46}$ | $26.61_{\pm3.88}$ | $29.15_{\pm2.91}$ |
| TT | GTrans | — | — | — | — | $30.19_{\pm4.37}$ |
| Lin | GraphAny (C) | — | — | — | — | $31.47_{\pm2.58}$ |
| | GraphAny (W) | — | — | — | — | $30.91_{\pm1.76}$ |
| SSL+FT | LP | $17.94_{\pm6.06}$ | $18.84_{\pm5.81}$ | $22.74_{\pm2.88}$ | $26.70_{\pm1.65}$ | $31.59_{\pm1.89}$ |
| | DGI | $18.06_{\pm6.22}$ | $18.95_{\pm5.99}$ | $21.98_{\pm4.07}$ | $25.84_{\pm2.76}$ | $28.81_{\pm2.43}$ |
| | GCL | $15.53_{\pm8.89}$ | $17.66_{\pm7.44}$ | $19.95_{\pm5.89}$ | $23.47_{\pm4.43}$ | $26.08_{\pm2.69}$ |
| SSL+Prompt | GPPT | $19.96_{\pm7.63}$ | $20.15_{\pm4.39}$ | $24.40_{\pm2.75}$ | $24.98_{\pm2.56}$ | $25.80_{\pm2.06}$ |
| | GPF | $20.89_{\pm8.20}$ | $24.77_{\pm6.08}$ | $27.94_{\pm6.10}$ | $28.45_{\pm4.01}$ | $29.19_{\pm3.67}$ |
| | GraphPrompt | $22.61_{\pm4.66}$ | $23.72_{\pm3.89}$ | $26.50_{\pm2.77}$ | $27.89_{\pm3.25}$ | $29.03_{\pm2.70}$ |
| | ProNoG | OOM | OOM | OOM | OOM | OOM |
| | DAGPrompt | $23.08_{\pm8.14}$ | $25.31_{\pm5.82}$ | $29.79_{\pm3.14}$ | $30.36_{\pm2.09}$ | $31.17_{\pm3.09}$ |
| | All-In-One | $15.29_{\pm7.55}$ | $19.78_{\pm5.50}$ | $24.06_{\pm3.38}$ | $28.89_{\pm2.07}$ | $29.01_{\pm3.98}$ |
| GFMs | GCOPE | $17.98_{\pm5.51}$ | $21.04_{\pm4.89}$ | $25.75_{\pm3.06}$ | $30.09_{\pm1.82}$ | $31.15_{\pm2.48}$ |
| | RiemannGFM | OOM | OOM | OOM | OOM | OOM |
| | MDGPT | OOM | OOM | OOM | OOM | OOM |
| | SAMGPT | OOM | OOM | OOM | OOM | OOM |
| | **GFMate** | **$30.19_{\pm3.65}$** | **$30.61_{\pm2.97}$** | **$31.36_{\pm2.90}$** | **$33.80_{\pm2.55}$** | **$34.47_{\pm2.69}$** |

*Table 21.* **Cross-domain transfer learning performance of graph classification under one-shot and three-shot settings.** Each dataset contains two columns corresponding to one-shot and three-shot results. The average accuracy (%) over five runs with standard deviation is reported. **Best** and Runner-up results are highlighted.

| | Dataset | PROTEINS | | COX2 | | BZR | |
|---|---|---|---|---|---|---|---|
| | Shot | 1-shot | 3-shot | 1-shot | 3-shot | 1-shot | 3-shot |
| SL | GCN | $51.24_{\pm7.96}$ | $53.89_{\pm3.52}$ | $45.59_{\pm9.62}$ | $59.74_{\pm8.96}$ | $44.89_{\pm16.27}$ | $52.65_{\pm9.06}$ |
| | GAT | $50.89_{\pm8.04}$ | $52.53_{\pm4.60}$ | $50.71_{\pm7.26}$ | $62.29_{\pm8.47}$ | $46.58_{\pm15.83}$ | $55.34_{\pm9.22}$ |
| | SAGE | $51.39_{\pm6.92}$ | $53.58_{\pm3.96}$ | $55.74_{\pm5.98}$ | $63.36_{\pm6.52}$ | $52.39_{\pm12.98}$ | $58.85_{\pm7.63}$ |
| | GPR | $51.09_{\pm9.26}$ | $52.90_{\pm5.05}$ | $48.15_{\pm9.69}$ | $58.73_{\pm7.99}$ | $45.16_{\pm17.63}$ | $53.06_{\pm8.29}$ |
| | H2GCN | $52.28_{\pm8.75}$ | $54.41_{\pm3.69}$ | $53.67_{\pm7.82}$ | $60.08_{\pm7.81}$ | $49.88_{\pm13.35}$ | $54.40_{\pm8.81}$ |
| SSL+FT | LP | $52.29_{\pm7.50}$ | $53.92_{\pm3.57}$ | $53.91_{\pm12.24}$ | $65.49_{\pm10.16}$ | $52.29_{\pm16.73}$ | $55.57_{\pm9.68}$ |
| | DGI | $49.68_{\pm8.37}$ | $50.28_{\pm4.06}$ | $54.42_{\pm10.03}$ | $60.07_{\pm12.25}$ | $49.79_{\pm14.95}$ | $50.73_{\pm8.96}$ |
| | GCL | $50.75_{\pm7.86}$ | $51.17_{\pm4.43}$ | $46.61_{\pm9.85}$ | $58.84_{\pm10.75}$ | $50.48_{\pm16.89}$ | $53.59_{\pm9.57}$ |
| SSL+Prompt | GPF | $54.36_{\pm4.99}$ | $55.09_{\pm2.89}$ | $60.49_{\pm14.56}$ | $65.30_{\pm13.37}$ | $58.71_{\pm15.45}$ | $70.05_{\pm12.47}$ |
| | GraphPrompt | $55.79_{\pm9.62}$ | $55.82_{\pm3.04}$ | $62.13_{\pm9.38}$ | $65.76_{\pm11.82}$ | $55.07_{\pm13.18}$ | $58.89_{\pm9.92}$ |
| | ProNoG | $53.59_{\pm7.65}$ | $54.07_{\pm4.18}$ | $57.04_{\pm11.91}$ | $60.23_{\pm12.28}$ | $52.25_{\pm15.86}$ | $55.47_{\pm13.36}$ |
| | DAGPrompt | $56.57_{\pm4.84}$ | $57.45_{\pm3.96}$ | $56.28_{\pm8.34}$ | $63.39_{\pm11.75}$ | $55.47_{\pm17.58}$ | $65.69_{\pm9.31}$ |
| | All-In-One | $56.68_{\pm8.37}$ | $57.72_{\pm5.94}$ | $58.76_{\pm8.52}$ | $66.27_{\pm9.94}$ | $61.62_{\pm13.47}$ | $62.95_{\pm10.86}$ |
| GFMs | GCOPE | $53.21_{\pm6.44}$ | $55.58_{\pm4.45}$ | $54.90_{\pm13.25}$ | $62.29_{\pm10.71}$ | $58.15_{\pm16.09}$ | $60.08_{\pm11.70}$ |
| | MDGPT | $54.70_{\pm6.69}$ | $55.06_{\pm6.49}$ | $50.47_{\pm12.35}$ | $59.81_{\pm11.85}$ | $53.48_{\pm12.92}$ | $55.90_{\pm12.81}$ |
| | SAMGPT | $55.19_{\pm5.52}$ | $55.27_{\pm5.86}$ | $52.07_{\pm7.82}$ | $63.37_{\pm9.38}$ | $58.59_{\pm17.04}$ | $66.35_{\pm14.07}$ |
| | **GFMate** | **$59.47_{\pm6.08}$** | **$61.24_{\pm3.62}$** | **$66.39_{\pm4.39}$** | **$69.82_{\pm1.18}$** | **$65.77_{\pm8.40}$** | **$77.06_{\pm2.18}$** |

# G. Discussion between pre-training-entangled and agnostic GFM prompts

The proposed pre-training-agnostic prompt differs from existing pre-training-entangled prompts in the following aspects.

**1. Conceptual perspectives**

- **Pre-training-entangled methods** in prior work prioritise knowledge specific to certain source domains and pre-train their prompts accordingly, under the assumption that the target domain is closely related to the source domain, as discussed in MDGPT (Yu et al., 2024c).

- **Pre-training-agnostic methods** prioritise information specific to the target domain and employ test-time prompt tuning directly on the target domain, obviating any requirement for prompt pre-training. This approach is particularly appropriate in GFM cross-domain scenarios, where the target domain is unseen during pre-training and may exhibit a distribution distinct from the source domains.

**2. Technical perspectives**

- Prompts in **pre-training-entangled methods** are optimised jointly with a fixed backbone and a fixed pre-training objective and require retraining whenever a novel training domain is introduced.

- Prompts in **pre-training-agnostic methods** are learned entirely at test time, without any pre-training, and without dependence on a specific backbone architecture or pre-training strategy. This design ensures compatibility with a broad range of backbones and GFMs employing diverse pre-training procedures, as demonstrated in Section 4.4 and Table 3.

**3. Advantages and Limitations**

- **Pre-training-entangled methods**:
  1. Effective utilisation of source domain information when the target domain is closely related,
  2. Limited generalisability when the target domain differs substantially,
  3. Reduced computational efficiency due to the requirement to retrain prompts for new domains,
  4. Constrained compatibility across different GFM architectures and backbone choices.

- **Pre-training-agnostic methods**:
  1. Enhanced capability to exploit the testing distribution of unseen target domains and improved generalisability across diverse GFM backbones,
  2. Potential computational overhead arising from test-time prompt tuning.

Section 4.2 further evaluates the efficiency of GFMate, demonstrating that it substantially outperforms existing GFMs relying on fine-tuning under a fair comparison at the GFM downstream adaptation stage. The centroid and layer prompt designs eliminate the need to train a separate prompt for each testing sample, in contrast to instance-level prompts in existing GFMs which must be trained individually for each node.

# H. Discussion between few-shot and test-time prompt tuning in GFM setting

The proposed test-time graph prompt tuning is fundamentally distinct from the existing few-shot prompt tuning task, as discussed in Remarks in the main paper. This distinction primarily arises from how the abundant testing data is utilised.

- **Existing Few-Shot Prompt Tuning for GFMs (e.g., SAMGPT and MDGPT (Yu et al., 2024c; 2025)**
  - Operate in a transductive setting where abundant testing data is accessible. This testing information is utilised **passively** as neighbourhood context when encoding the few labelled nodes.
  - The substantial distribution gap between the limited labelled samples and the abundant unlabelled testing samples remains unaddressed.

- **Proposed Test-Time Prompt Tuning for GFMs**

  - Actively leverages the accessible testing samples, which previous methods overlook. This active utilisation enables the prompts to adapt effectively to the testing data in the unseen target domain, thereby enhancing the generalisation performance of the GFM.
  - Uses target domain data to **directly optimise the prompts during inference**, without reliance on prompt pre-training or model pre-training strategies.

## I. Limitations

One limitation of GFMate lies in its design for GNN-based GFMs (Zhao et al., 2024a; Yu et al., 2024c; 2025; Sun et al., 2025; Wang et al., 2025), making it inapplicable to LLM-based GFMs (Liu et al., 2023a; Li et al., 2024; Chen et al., 2024a; Xia et al., 2024; Chen et al., 2024b; Kong et al., 2025), whose backbone models are fundamentally different. GFMate is tailored for graph data without textual attributes and thus cannot exploit text information present in text-attributed graphs. Developing test-time methods compatible with both GNN- and LLM-based GFMs would further enhance the generalisability of this work.

