# OpenReview forum: "GFMate: Empowering Graph Foundation Models with Test-time Prompt Tuning"
_ICML.cc/2026/Conference — ICML 2026 regular_

### Official Review · Reviewer_aA9H · 2026-02-26

**Soundness:** 2
**Presentation:** 2
**Significance:** 2
**Originality:** 2
**Overall Recommendation:** 4
**Confidence:** 5

**Summary:**

This paper proposes GFMate, a pre-training-agnostic test-time prompt tuning framework for graph foundation models that decouples prompt learning from specific source domains and pre-training strategies.
By introducing centroid and layer prompts alongside a test-time complementary learning objective, GFMate actively leverages both few-shot labelled and abundant unlabelled target data.

**Compliance With Llm Reviewing Policy:**

Affirmed.

**Final Justification:**

Thank you for the authors’ response. Overall, the motivation of the paper is reasonable, and the empirical evaluation is fairly comprehensive. Based on these clarifications, I have accordingly adjusted my score.

One key concern is that graph data across domains often exhibit significant heterogeneity: not only do feature spaces differ in dimensionality, but the semantics associated with each dimension can also vary. As a result, directly pre-training on such multi-domain data may lead to misaligned or even conflicting representations, rather than meaningful knowledge transfer.
The authors are encouraged to provide more discussion on how this issue is addressed in their framework.

**Key Questions For Authors:**

See above.

---
## After Rebuttal：

Thank you for the authors’ response. Overall, the motivation of the paper is reasonable, and the empirical evaluation is fairly comprehensive. Based on these clarifications, I have accordingly adjusted my score.

One key concern is that graph data across domains often exhibit significant heterogeneity: not only do feature spaces differ in dimensionality, but the semantics associated with each dimension can also vary. As a result, directly pre-training on such multi-domain data may lead to misaligned or even conflicting representations, rather than meaningful knowledge transfer.
The authors are encouraged to provide more discussion on how this issue is addressed in their framework.

**Limitations:**

The authors do not adequately discuss the limitations of the proposed method.

**Strengths And Weaknesses:**

## Pros
1. Graph foundation models  represent an emerging and highly active area of research.

2. The authors have conducted extensive and comprehensive experimental analyses.

## Cons

1. The core philosophy of graph prompting is to bridge the gap between pre-training objectives and downstream tasks, thereby unleashing or enhancing the encoder's performance on arbitrary downstream applications. However, the Introduction is poorly structured in this regard; it is difficult for readers to grasp the exact mechanism and underlying purpose of existing prompt-based methods. Furthermore, existing GFM-related works mentioned by the authors, such as MDGPT, SAMGPT, and MDGFM, do not appear to represent traditional prompting tasks. Their core objective is fundamentally to align distribution discrepancies across different source domains using learnable vectors. In this sense, these approaches and traditional prompting are entirely different concepts. The authors fail to adequately clarify this distinction or justify why they are directly comparable.

2. The claim regarding "(ii) Limited cross-model generalisability"  lacks sufficient scientific justification. The authors argue that pre-training-entangled prompts cannot be easily generalized across different pre-trained models. However, it is entirely unclear why generalizing prompts across different pre-trained models is necessary, or what specific benefits it yields. Achieving cross-model prompt transferability does not seem to align with the primary design goals of a GFM. The authors must elaborate on why this is considered a critical limitation and why existing works are fundamentally unable to achieve it.

3. The authors identify a challenge stating that current graph prompt tuning paradigms rely solely on few-shot samples while neglecting abundant un-labelled data despite their availability. This assertion is conceptually problematic. The fundamental premise of a foundation model is precisely to leverage minimal (few-shot) to perform in-context learning and generalize to unseen data. By criticizing the reliance on few-shot samples and requiring the active exploitation of target-domain unlabeled data for test-time adaptation, the authors' underlying assumption seemingly contradicts the core philosophy and intended deployment scenarios of foundation models.

4. Figure 1 fails to clearly convey the core insights and mechanisms of the proposed method. Additionally, the phrase "(iii) Vulnerability to source domain bias"  is highly ambiguous. It lacks sufficient explanation or context, making it extremely difficult for the reader to comprehend the specific issue being addressed or how the proposed framework effectively resolves it.

5. The authors do not provide a clear and detailed explanation of how the multi-source domain pre-training is executed within the proposed framework.

---

> ### Author Rebuttal · Authors · 2026-03-30
>
> **We sincerely thank you for your review. We have carefully addressed your concerns below:**
>
> >**Re W1: Why Compare to GFM Prompt But Not Traditional Graph Prompt**
>
> We kindly clarify that **our paper focuses on GFM prompting** (MDGPT, MDGFM, SAMGPT) for **multi-domain graph transfer learning**. We **do not** aim to touch anything **about traditional graph prompting** (GraphPrompt, GPPT, GPF), which works on **single-domain graph supervised learning**. GFM prompt is a more challenging and timely scenario compared with the supervised learning-based graph prompting. The techniques used in these two areas are totally different.
>
> From the title, abstract and introduction, **we all position our paper under GFM prompting**. We believe this positioning is **adequate and timely by focusing on the more challenging transfer learning scenario instead of supervised learning scenario**. **We kindly invite the reviewer to reconsider the position of our paper under GFM prompting.**
>
> >**Re W2: Necessity of Cross-model Generalisability**
>
> We kindly clarify that **a method that can generalise to different backbone/base models is a good merit**. For example, the prompt tuning in LLMs that can work under different variants of transformers is the cornerstone for the success of LLM's pretraining and finetuning.
>
> In **most existing GFMs**, they design their models and their corresponding prompts. These prompts are not generalisable across different GFM backbones, which is exactly the "entanglement" claim in our paper. **If a new backbone comes out, the corresponding GFM prompt needs a brand-new design**.
>
> As in Introduction Line 65, cross-model generalisability is highly **necessary to avoid redevelopment of prompts for different GFMs.** Our GFMate can be seamlessly applied to **various backbones and GFMs, as in App E.2 and Tab 6, App E.3 and Tab 7.**
>
> >**Re W3: Claim on Unlabelled Data:**
>
> We kindly clarify that **both GFMate and existing GFM utilise the same amount of information: labelled few-shot data and unlabelled data**. As discussed in Remark 2 (Line 126) and introduction (Line 56&79) for **most existing GFMs** operate **in a transductive setting, where the unlabelled testing nodes are already available and utilised together with the labelled few-shot nodes.** These labelled and unlabelled nodes are all encoded with GNNs under the message passing mechanism. Thus, **the application scenarios are strictly consistent with existing GFMs**.
>
> As in Line 77, for existing GFMs, **the abundant unlabelled nodes from target domain are available during prompt tuning but are only being exploited passively as neighbour contexts**. Therefore, we do not require extra information than other GFMs like MDGFM.
>
> >**Re W4.1: Fig 1 Core Insight & Mechanism**
>
> We apologise for causing confusion in Fig. 1 without sufficient information.
>
> 1. **Core Insight** of our paper is the **unlabelled nodes not being actively learnt during prompt tuning, although these nodes already present**.
>
> * **Figure 1(a)** shows **grey unlabelled nodes in the target Social domain are not actively used, even they are already present** during few-shot prompt tuning.
>
> 2. The main **mechanism** of our GFMate is **(1) design pre-train-agnostic prompts** and **(2) test-time prompt tuning with both few-shot and unlabelled nodes**.
>
> * **Figure 1(b)** shows that **(1) prompts in red are not involved during pre-training** and (2) **test-time graph can also guide prompt learning (red)**.
>
> Follow your suggestion, we **have updated a clearer version with sufficient legend information** in our paper.
>
> >**Re W4.2: Source Domain Bias**
>
> We apologise for causing ambiguity.
>
> The vulnerability of source domain bias is that **it's possible that one or some domains may dominate the multi-domain pre-training** due to potential imbalanced number of source domain data, task difficulty, etc. This would **present data bias in the prompt training** if it's pre-training-entangled like prior GFM prompts.
>
> We have updated this part as:
> Existing GFM prompts are pre-trained on multiple source domains. During pre-training, dominant source domains could suppress learning from others, leading to overfitting and limited generalisation to other domains.
>
> >**Re W5: Pre-training**
>
> Apologies for any missing details. As **in Line 191, we follow existing GFMs** (MDGPT, etc.) to use **link prediction as the pre-training task**. Specifically, we sample edges from each source domain and jointly pre-train models by link prediction.
>
> As a pre-training-agnostic method, we do **not claim contribution on pre-training** and we aim to develop **GFMate applicable to various pre-training methods.** In **App E.3 and Tab 7, we apply GFMate prompting on different pre-trained GFMs**, and GFMate achieves **significant improvement** on all pre-training methods.
>
> **We appreciate your highly professional questions, which greatly strengthened our manuscript.** If you would consider raising the score, we would be very grateful.

---

> > ### Author Rebuttal · Reviewer_aA9H · 2026-04-03
> >
> > Thank you for the authors’ response. I attempted to submit an official comment yesterday, but it seems the authors may not be able to see it. Therefore, I reproduce my comments here:
> >
> > First, I understand the authors’ original intention: they aim to address the problem of how to transfer tokens learned from source domains to downstream tasks. However, the current presentation and motivation, method are somewhat confusing.
> >
> > Specifically, in prior works such as SAMGPT and MDGFM, these components are consistently referred to as tokens. In this paper, however, the authors use the term prompts, which is somewhat misleading. The concept of prompting is generally introduced to bridge the gap between pre-training and downstream tasks, rather than to address issues specific to GFM. In contrast, the core challenge of GFM lies in learning transferable representations from heterogeneous, multi-source graph data.
> >
> > Moreover, the pre-training procedure adopted in this paper does not introduce any fundamentally new mechanism. It closely resembles existing self-supervised graph learning approaches such as GraphCL or GraphMAE, with the main difference being the use of a larger collection of datasets. However, pre-training on multiple heterogeneous graph domains does not necessarily guarantee better representation learning; due to significant distributional differences across domains, the model may even struggle to learn meaningful shared knowledge. Therefore, the reported performance gains may not stem from improvements in the GFM.
> >
> > Finally, the paper claims as Challenge 2 that “current graph prompt tuning paradigms rely solely on few-shot data while neglecting abundant unlabeled samples.” This claim appears questionable. In inductive settings, such unlabeled target-domain data may not always be accessible, making the comparison potentially unfair. If no additional target data are available, the proposed method would not be applicable. Furthermore, prior work such as SAMGPT already suggests that their pre-training framework is flexible and not tied to a specific setting.

---

> > > ### Author Response · Authors · 2026-04-03
> > >
> > > **Thank you for your valuable follow-up, which largely helps clear up this conversation.**
> > >
> > > > **Re Terminology of “prompt” vs “token”:**
> > >
> > > **In GFM, prompt and token are the same in most literature**. Both *prompt* and *token* are essentially just learnable embedding vectors as in prior GFM papers. In **GFMate**, our prompts are also learnable embeddings that can be composed via addition and multiplication, and thus we safely adopt the term *prompt* defined in Remark 2, consistent with both SAMGPT and MDGFM.
> > >
> > > >**Re GFM Prompt, Traditional Graph Prompt and Pre-train/Downstream Task**
> > >
> > > Our paper focuses on **GFM prompting** (e.g., SAMGPT, MDGFM), **trained on multiple seen domains and tested on multiple unseen domains**. These **GFM prompting is bridging the gap** mainly in pre-train and downstream domains.
> > >
> > > **Traditional graph prompting** (e.g., GPPT, GPF) is **trained and tested in the same dataset/domain**, similar to supervised learning. They are **not trying to bridge the gap** because they operate on a **same task/dataset**.
> > >
> > > This is also supported by **prior GFM work MDGFM (ICML2025)**:
> > > - “Additionally, we develop an efficient prompt learning strategy to transfer knowledge from multiple source domains to a target domain.” (Page 2, Abstract)
> > > - “the learnable specific prompt $p_s \in \mathbb{R}^d$ aims to directly align the target domain with the unified semantic space” (Page 5, Sec. 4.3)
> > >
> > > We fully agree that "the core challenge of GFM lies in learning transferable representations from heterogeneous, multi-source graph data". This is **exactly what we and other GFM did**. We kindly redirect you to our formal Definition 1 on this issue.
> > >
> > > > **Re Pre-training Procedure**
> > >
> > > 1. **We did not claim any novelty/tech contribution in pre-training. Our main contribution is in how to improve transfer learning in GFMs by pre-training-agnostic prompts, which are not pre-trained**.
> > >
> > > 2. **All GFMs are pre-trained on multiple datasets. And no GFM will claim novelty in just pre-trained on multiple datasets**.
> > > * As you mentioned **GraphCL and GraphMAE, these are pre-training methods on a single domain**, and they **don't focus on cross-domain transfer**. We also have experiments (Tab 1) with comparison with GraphCL (GCL+FT), which shows a large performance gap, even when we put GraphCL in a favourable supervised learning setting.
> > > * **GFMs focus on the transfer learning from seen datasets to unseen datasets**. Different focus, different contributions.
> > >
> > > 3. GFMs are focusing on how to **utilise the rich information in multiple heterogeneous graph domains to help with the generalisation to other unseen domains**.
> > > * The results from **existing GFMs show a clear improvement in using multiple pre-trained datasets**.
> > > * Our **GFMate uses the same GFM setting** (multiple seen domains for pre-training and multiple unseen domains for testing) under **the same and simple pre-training** strategy, but with a **different prompting and generalisation** design.
> > > * This ablation **clearly verifies that the reported performance gains stem from improvements in the GFM**.
> > >
> > > > **Re Inductive Setting**
> > >
> > > * **We analyse in inductive settings in App E.10 and Table 12**, where no unlabelled test nodes are available. **GFMate still consistently improves over SOTA GFM**. More evidence on fair comparison is in Response to Reviewer rSak W3.
> > >
> > > > **Re SAMGPT's Prompt Pre-trained Flexibility**
> > >
> > > We **totally agree with you that the pre-training task is flexible for most GFMs**, contrastive learning, link prediction, etc.
> > >
> > > **But more importantly, the prompt/token design of most existing GFMs is not flexible and is tied to the pre-training or model layer.** SAMGPT stated “we inject a series of structure tokens into the graph encoder, modifying the structure-based aggregation at each layer” (Page 4, Sec 4), showing that **its prompt design is explicitly coupled with the pre-training phase and the graph encoder (a 3-layer GCN).**
> > >
> > > **Our prompt is flexibly applied to different GFMs and backbones as in the experiment of Table 7 in App E.3 and Table 6 in App E.2**, supporting the flexibility of GFMate.
> > >
> > > **Thank you again for the very valuable questions. If our response has still not addressed your concerns, please also kindly let us know.**
> > >
> > > **——————————————————————— Update 7 Apr ————————————————————————**
> > >
> > > **We sincerely thank the reviewer for raising the score and providing the final justification in support of our work.**
> > >
> > > Following your valuable suggestions, we have further included a detailed discussion of the potential limitations in multi-domain pre-training, particularly the issue of heterogeneous feature spaces and possible representation misalignment in our manuscript. This makes our manuscript more comprehensive and complete.
> > >
> > > **We wholeheartedly thank you again for recognising that the motivation of the paper is reasonable and that the empirical evaluation is comprehensive. We would be most grateful for your support during the upcoming AC discussion phase.**

---

### Official Review · Reviewer_JKN9 · 2026-03-06

**Soundness:** 2
**Presentation:** 2
**Significance:** 2
**Originality:** 3
**Overall Recommendation:** 3
**Confidence:** 4

**Summary:**

This paper proposes GFMate, a pre-training-agnostic test-time prompt tuning framework designed to enhance the performance of GFMs in few-shot cross-domain scenarios. The core innovations of the method are: 1) centroid prompts and layer prompts; 2) a test-time graph complementary learning objective.

**Compliance With Llm Reviewing Policy:**

Affirmed.

**Final Justification:**

I have reviewed the authors’ response and decided to keep my original score.

**Key Questions For Authors:**

1.What is the basis for the statement in lines 196-197 that "provides possibility for centroids to move towards the real centre of a cluster in target domains"?
What is the basis for the statement in lines 239-240 that "ensembling the layer-wise predictions corresponding to different hop-aggregated representations to improve the GFM adaptation on the unseen target domain"?

**Limitations:**

I only found a discussion on the limitations of pre-training-agnostic methods. Adding a limitations section for GFMate would help round out the work. Apart from the additional computational costs, are there any limitations that might be encountered when using the GFMate framework in practical applications such as social recommendation and molecular drug synthesis?

**Strengths And Weaknesses:**

Strengths:

1.GFM is a important research direction in graph learning, and the problem this paper tackles is meaningful.

2.The experiments are comprehensive and the results are strong.

3.The paper is well-written, clear, and easy to follow.

Weaknesses

1.The core motivation of the paper raises some concerns. The notion of "prompt entanglement" mentioned in lines 18-24 of the abstract seems to conflict with the widely adopted "pre-training and fine-tune" paradigm typically associated with GFMs. The claim "restricts their generalisability to other domains and different GFMs" requires more rigorous justification, such as theoretical analysis or experimental validation, to be convincing.

2.In line 138, "Existing prompts assume the source domain informs the target domain." I think this assumption is actually fundamental to the idea of foundation models in general—if the source and target domain distributions are too far apart, pre-training itself becomes meaningless, and prompt tuning would be unlikely to help. This point needs to be addressed more carefully.

3.In line 153, "neglect of unlabelled target domain data" needs further clarification. I'm not entirely convinced this critique holds universally across both node-level and graph-level tasks. Prompt tuning is typically designed to work with limited labeled data, which is a widely accepted and reasonable setup. Incorporating unlabeled data would inevitably introduce additional training overhead, and the potential performance gains may be marginal.

4.Cross-domain generalization is inherently challenging, and the proposed GFMate framework is a commendable attempt in this direction. That said, the core design choices behind GFMate for handling cross-domain tasks need to be better motivated. See Key Questions For Authors.

---

> ### Author Rebuttal · Authors · 2026-03-29
>
> **We sincerely appreciate your constructive feedback. We have thoroughly addressed your questions and summarise:**
>
> >**Re W1: Motivation and Justification of Entangled Prompt**
>
> * **Motivation**: We kindly clarify that **prompt entanglement refers specifically to GFM prompts rather than GFM backbone pre-training.** As in Lines 47 & 133, existing prompts are often encoded directly with a specific GFM pre-training strategy or coupled with a set of pre-defined source domains.
>
> * **Justification**: In **Fig 3, we empirically observe that baseline GFM prompts entangled with pretraining do not transfer well in target domains**, which supports **generalisability restriction**. More discussions are in App G.
>
> >**Re W2: Prompt Cross-donaim Assumption**
>
> We fully agree with you on the carefulness on this point. We will **have a more conservative tone** on the description. We have **updated our manuscript** for this point as:
>
> "Existing prompts generally assume the source domain could inform the target domain meaningfully. Yet this assumption would fail when the unseen target domain has a significantly different distribution, where the pre-trained prompts may have been deeply entangled with source domains."
>
> >**Re W3: GFM Prompt with Unlabelled Data**
>
> * **Motivation**: (1) **Existing GFM prompts** use **both limited labelled nodes for supervision signal and abundant unlabelled nodes** to support message passing and neighbour aggregation in a transductive setting. (2) Our **GFMate use the same amount of information fairly**. We develop additional training objectives to utilise the abundant unlabelled data more meaningfully. We have thorough descriptions (Line 70-85) and visualisation (Fig 1a).
>
> * **Training Overhead**: In **Sec 4.2 & Fig 5, GFMate has the highest efficiency in training convergence time and GPU memory** compared with other GFMs. This high efficiency is primarily driven by **test-time complementary learning providing extra meaningful signals**.
>
> * **Potential Marginal Gain**: In **Tab 1 & 14 for overall and different-shot experiment, GFMate shows non-marginal improvement** with up to 30.63% over SOTA GFMs/traditional prompts/supervised methods.
>
> >**Re W4&Q1.1: Centroid Moving in Line 196-197**
>
> Our learnable centroid design is to **learn a vector change $\beta$ (Eq. 4) that adds to the fixed centroid**. With the labelled and unlabelled target domain data, we can expect the centroid to move along the direction given by the learned vector change $\beta$, towards the real centre that gives the maximum classification performance. This can be further verified by **visualisations of the centroid movement in Fig 8, App. E.8, where centroids move closer to class centres after GFMate**.
>
> Most existing GFMs perform classification by comparing test embeddings with the few-shot embedding's centroids. **These existing centroids are fixed as the few-shot embeddings, and may deviate from the true test centres, as visualised in Fig 3.**
>
> >**Re W4&Q1.2: Layer-wise Ensemble in Line 239-240**
>
> Our design of a layer-wise ensemble for effective target transfer is based on
> 1. The **pre-trained backbone GNN generally has a fixed structure (e.g., fixed number of layers)**;
> 2. The **target domains are different from pre-trained domains**, so that directly using the information from the **pre-trained GNN would be sub-optimal**.
>
> Our design aims to
> 1. Let **different layers contribute to the final prediction differently**;
> 2. Let this **contribution adapt to different domains with varying neighbour patterns**. For example, when transferring to both homophilic and heterophilic graphs, our **learnable layer-wise ensemble can adapt to these graphs differently by using neighbouring information from layers differently**.
>
> **Visualisations** of different predictions for different layers on target domains are provided in **Fig 2. The pre-trained model's performance on target domains varies across layers**, and relying on a single-layer prediction is suboptimal. A learned ensemble of layer-wise predictions can improve GFM adaptation compared to fixed, pre-defined layer predictions.
>
> >**Re L1.1: Add Limitations**
>
> **Following your suggestion, we have updated our manuscript to include a limitation section** on incompatibility with LLM-based backbones in addition to App. G. Generally, GFMate is designed for GNN-based GFM following MDGFM, and is not applicable to LLM-based backbones.
>
> >**Re L1.2: Computation Cost and Limitations on RecSys Graph or Molecure**
>
> We have **evaluation of GFMate on the recommendation dataset (Amazon Photo, Tab1) and molecular datasets (BZR, COX2 and PROTEINS, App F.2, Tab 19)**. We would like to emphasise again that **our training is actually more efficient as in W3. GFMate is also more effective** compared with baselines for these datasets.
>
> **We greatly appreciate your insightful questions, which have significantly improved our manuscript.** If you would consider raising the score, we would be very grateful.

---

> > ### Author Rebuttal · Reviewer_JKN9 · 2026-04-03
> >
> > Thank you for your rebuttal. I'd like to maintain my score.

---

> > > ### Author Response · Authors · 2026-04-03
> > >
> > > **We sincerely thank you for considering our rebuttal.**
> > >
> > > We note that several key concerns raised in the original review **(including the motivation of prompt entanglement, cross-domain assumptions, and the technical justifications for centroid learning and layer-wise ensembling) have been addressed with additional explanations and supporting evidence in our rebuttal.**
> > >
> > > As the score has been maintained without further clarification, it is currently unclear which specific points may still be considered insufficiently addressed.
> > >
> > > To facilitate constructive revision, **we would greatly appreciate it if you could kindly indicate any specific remaining concerns that contributed to maintaining the current score. We are eager to address any of your remaining concerns so that we can further improve our paper.**
> > >
> > > We also hope this clarification may assist the AC in evaluating whether the main concerns have been sufficiently addressed and whether the current evaluation reflects the updated manuscript and the discussion.
> > >
> > > **We wholeheartedly thank you again for your time and consideration.**
> > >
> > > Sincerely,
> > >
> > > Authors 2558

---

### Official Review · Reviewer_1QRX · 2026-03-08

**Soundness:** 3
**Presentation:** 4
**Significance:** 3
**Originality:** 3
**Overall Recommendation:** 5
**Confidence:** 2

**Summary:**

The authors point out the limitations of current graph prompt tuning on graph-based models: the prompt words are deeply coupled with the pre-training task/source domain, and the value of unlabeled data in the test set is ignored. The GFMate framework is proposed, introducing "Centroid Prompt" and "Layer Prompt", and combining it with the goal of Complementary Learning at Test Time to achieve pre-training-independent prompt fine-tuning that can utilize test data.

**Compliance With Llm Reviewing Policy:**

Affirmed.

**Final Justification:**

The rebuttal adequately addressed my main concerns regarding complementary label stability and cross-model generalization, providing convincing empirical evidence in support. I maintain my positive assessment and recommend acceptance.

**Key Questions For Authors:**

1. Are the complementary tags stable in the initial stages of test-time optimization? In case the complementary tags are noisy, then the optimization task may result in instability or incorrect gradients during training.
2. Can the authors provide other evidence to show cross-model generalization like between-model generalization, by applying GFMate to more GFMs of other architectures?

**Limitations:**

Yes

**Strengths And Weaknesses:**

## Strengths
1. Rigorous motivation and extensive experiments: The authors first demonstrate two key points through empirical observation: first, the contribution of different layers of GFM to downstream tasks varies greatly, and second, few-shot nodes cannot represent the overall distribution of the test set. Based on this motivation, the "Layer Prompt" and "TGCL" are logically supported.
2. High quality of expression: The paper has intuitive and logical problem definitions and diagrams. The full text is rigorously structured, from challenge identification to methodological derivation to experimental demonstration, the narrative is fluent, and the formula definition is accurate.
3. Practical value: The proposed paradigm effectively decouples on-the-fly adjustments from specific pre-training strategies and source domains. This allows it to be an enhancer for various graph foundation models, allowing them to adapt to completely invisible target domains during inference.

## Weaknesses
1. Inadequate efficiency analysis: Since GFMate requires optimization of the process during the inference phase, it inevitably adds additional latency compared to normal zero-shot or fixed-prompt inference. While the author discusses complexity, a more detailed analysis of the efficiency of reasoning is also important.
2. Potential instability: The thesis assumes that the "lowest probability class" is a negative signal for safety. However, in cases where the distribution is offset and the GFM prediction is nearly uniform, these "complementary labels" can become noisy.

---

> ### Author Rebuttal · Authors · 2026-03-29
>
> **We sincerely thank you for your insightful questions.** We have thoroughly addressed your concerns and summarise our response with **highlighted** keywords as follows:
>
> >**Re W1: Efficiency and Latency**
>
> We fully agree with you to analyse the efficiency. We kindly invite you to look at **Section 4.2 and Figure 5** in our manuscript, where we already **provide a detailed efficiency analysis. GFMate achieves the highest efficiency** among existing GFMs under the same downstream prompt tuning stage.
>
> This high efficiency gain is primarily driven by our active exploitation of unlabelled data through test-time complementary learning, which ensures rapid convergence of the prompt tuning, while other methods, such as RiemannGFM and SAMGPT, require extensive fine-tuning to achieve the best performance during the same downstream stage. Therefore, **GFMate does not introduce additional test-time latency compared to normal few-shot prompt tuning in practice.**
>
>
> >**Re W2&Q1: Stability of Complementary Learning**
>
> The complementary labels in the initial stage of GFMate are stable and highly accurate. As in **Appendix E.11 and Table 13**, we compare the **accuracy of the predicted least similar class against our complementary labels** derived from the layer-wise entropy-based strategy in the first stage before prompt tuning, and observe that these **complementary labels are highly accurate and stable across datasets.** This can be attributed to our layer-wise entropy-based strategy, which is **motivated by the empirical observations in Figure 2.** In Appendix E.6 and E.7, we further demonstrate that GFMate, built upon these complementary labels, is robust under extreme noise cases, further verifying its stability.
>
> >**Re Q2: Cross-model Generalisation**
>
> Thank you for the valuable suggestion.
>
> 1. We provide experiments in **Appendix E.2 and Table 6 for generalisation analysis for cross-model**. We evaluate GFMate over **three additional model architectures, GraphSAGE, GAT** (homophilic backbones), and **H2GCN** (heterophilic backbone).
>
> 2. Additionally, in **Appendix E.3 and Table 7, we show GFMate's strong generalisability by plugging GFMate into models pre-trained by different objectives (e.g., DGI, GCL) and various GFMs (e.g., SAMGPT).** GFMate consistently empowers and improves the performance of all these diverse GFMs, which solidly confirms its strong cross-model generalisability.
>
> **We greatly appreciate your valuable suggestions and positive evaluation. We would be most grateful if you could champion our paper.**

---

> > ### Author Rebuttal · Reviewer_1QRX · 2026-04-01
> >
> > Most of the issues were resolved and I maintain my high score.

---

> > > ### Author Response · Authors · 2026-04-01
> > >
> > > **We wholeheartedly appreciate your highly professional review and recognition of our work. Your review provides an exceptionally precise and insightful evaluation of our work.**
> > >
> > > We would be most grateful for your support during the upcoming AC discussion phase.
> > >
> > > Sincerely,
> > >
> > > Authors 2558

---

### Official Review · Reviewer_rSak · 2026-03-09

**Soundness:** 3
**Presentation:** 3
**Significance:** 3
**Originality:** 3
**Overall Recommendation:** 5
**Confidence:** 3

**Summary:**

The authors observed that existing Graph Foundation Models struggles to generalize to new domains as their prompts are tied to the model during training. To solve this, they formalize the idea of "pre-training-agnostic" with a framework called GFMate, which adapts to new graphs purely during inference without touching the original pre-trained model. The system proposes test-time complementary learning, where they utilize unlabeled data to refine the class centroids, which is underexplored by existing GFMs. The empirical results are superior.

**Compliance With Llm Reviewing Policy:**

Affirmed.

**Final Justification:**

The author provides detailed responses, which addressed my concern during initial reviewing.

**Key Questions For Authors:**

NA

**Limitations:**

yes

**Strengths And Weaknesses:**

## Strength

The intuition is very interesting, and directly target the entanglement and test-data under-utilization problem. The experimental results shows great improvement, which validate the initiative, and I think this shed lights on next step to advance GFM. The theory is also clean and explain the effect of the loss well.

## Weakness
- I am concerned about completely decouple prompting from pre-training. This has two implications, 1, the framework is essentially transductive, always needs training when a new data emerges. 2, if the pre-training is not adaptive, meaning pre-training can not adapt in  message passing, and modeling heterophily and homophily will become difficult, I presume this is difficult to corrent for test-time training.

- The application scenario could be limited. For this framework to work well, we cannot have too few data, for example, if we are short on unlabelled test node, it will not be effective for many techniques mentioned in the paper. We also cannot have too much data, if so, we can simply go for a supervised model. So the method seems to be specifically designed for this case, limiting is utility.

- The main results can be improved. The comparison between GFMate and baselines is a bit apple-to-orange. As GFMate enjoy active gradient update from test nodes, other baseline is not utilize this information at all. On one hand, this is the architectural advantage of the proposal, on the other hand the technical advantage of the method is not clear, the author should compare to simpler variant that uses exactly the same amount of information.

- A major claim from the paper is this method is also model agnostic. However, the SVD based feature alignement does not adapt to features with vastly different dimensions. And this method can inherit the same problem, and hence model agnostic can be a bit over-claiming

---

> ### Author Rebuttal · Authors · 2026-03-29
>
> **We sincerely thank you for your valuable feedback,** and we have thoroughly addressed your concerns. We summarise our response and **highlight** the key information as follows:
>
> >**Re W1.1: GFM Prompt Training When New Data Emerges**
>
> We kindly clarify that **all current GFM prompt methods require prompt training when data from new domains emerges**, e.g., SAMGPT and MDGFM. Our **GFMate does not have an extra disadvantage compared with other GFMs in dealing with new data**. Even in the traditional graph prompt methods (e.g., GraphPrompt) that only for single-domain data, they still require fine-tuning after pre-training.
>
> >**Re W1.2: Message Passing, Heterophily and Homophily**
>
> Our design of **GFMate provides flexibility to adapt the message passing in test time tuning, when new data comes in**. This design actually releases the constraint of the pre-training coupled prompt in other methods in encoding too much information during pre-training. To evaluate this, we conduct **experiments in a wide range of both homophilic (Cora, Citeseer, Photo) and heterophilic (Texas, Cornell, Wisconsin, Chameleon, Squirrel) datasets as in Table 1**, where GFMate shows significant improvement in both homophilic and heterophilic scenarios. It verifies that **GFMate actually has a more effective prompt tuning than other GFMs in diverse domains**.
>
> >**Re W2.1: Results on Only Few-shot, No Unlabelled Data**
>
> We agree that this scenario is important, so that we have conducted experiments already. In **Appendix E.5 and Table 9, we conduct experiments in different percentages of unlabelled test nodes from 0%~100%**. Additionally, in **Appendix E.10 and Table 12, we conduct experiments in fully inductive scenarios, where no unlabelled test nodes are available**. GFMate is effective and achieves improved results compared with baselines in both experiments.
>
> >**Re W2.2: Results on All Data Labelled**
>
> We fully agree with you on this scenario, so that we have conducted experiments already in the paper. In **Appendix F.1 and F.2 (Table 14 to 18), we conduct experiments on full shot where all the available data is labelled**. GFMate remains powerful in performance compared with baselines.
>
>
> >**Re W3: Fair Comparison and Usage of Unlabelled Data**
>
> We respectfully clarify that **both GFMate and existing GFMS utilise the same amount of information: labelled few-shot data and unlabelled data**, ensuring a fair comparison. As discussed in Remark 2 (Line 126) and the introduction (Line 56 and 79) for **existing GFMs**, such as SAMGPT and MDGPT, they operate **in a transductive setting, where the unlabelled testing nodes are already available and utilised together with the labelled few-shot nodes.** These labelled and unlabelled nodes are all encoded with different GNNs under the message passing mechanism. Thus, **the information usage is fair between GFMate and existing GFMs**.
>
> Nonetheless, **we fully agree with you to conduct an experiment with different usage** of these unlabelled nodes for a more thorough comparison. In **Appendix E.10, we further compare GFMate with pseudo labelling methods of different percentages of unlabelled nodes**. Results show that GFMate consistently outperforms these strong variants **by a large margin.**
>
> >**Re W4: Model Agnostic and SVD**
>
> We kindly clarify that **(1)** the **model-agnostic** contribution of GFMate refers to **its prompt tuning without interfering with pre-training;** and **(2) we follow most existing baselines to use SVD projection, where we did not intend to claim any contribution.**
>
> 1. As discussed in Line 63 (Introduction) and Line 142, existing GFM prompting methods are typically coupled with specific model architectures and pre-training strategies, limiting their ability to generalise across different models. For example, the domain prompts in SAMGPT are jointly trained with a GCN model via graph contrastive learning and cannot be easily transferred to other pre-trained models, such as GAT. In contrast, GFMate can be applied across different models and pre-training methods **(Appendix E.2, E.3). Therefore, GFMate is model-agnostic owing to its plug-and-play ability.**
>
> 2. We strictly follow the **same SVD-based projection** used in prior baselines (e.g., SAMGPT) to ensure a **rigorously fair comparison under consistent and identical feature projection settings** for all source and target domain data (Line 187). **Our contribution is not in SVD.**
>
> **We sincerely thank you for your constructive feedback, encouraging suggestions and positive evaluation, which have greatly helped improve our manuscript.** If you would consider raising the score, we will be really grateful.

---

> > ### Author Rebuttal · Reviewer_rSak · 2026-04-03
> >
> > Thanks for the rebuttal, my concerns are addressed.

---

> > > ### Author Response · Authors · 2026-04-03
> > >
> > > **We sincerely appreciate your professional and insightful review of our work. Your comments provide a clear and valuable evaluation that greatly helps us improve the paper.**
> > >
> > > We would be most grateful for your kind support during the upcoming AC discussion phase.
> > >
> > > Sincerely,
> > >
> > > Authors 2558

---

### Decision · Program_Chairs · 2026-04-30

**Decision:**

Accept (regular)

**Comment:**

This paper introduces GFMate, a pre-training-agnostic test-time prompt tuning framework for graph foundation models. The work is well-motivated, clearly presented, and addresses an important limitation in current graph prompting paradigms. Empirical results are strong and supported by thorough experiments and analysis. While concerns remain regarding efficiency, assumptions about prompting, and applicability under extreme data regimes, reviewers are generally positive about this work and believe these issues can be addressed in a revised version.